# Graph Neural Preconditioners for Iterative Solutions of Sparse Linear Systems

**Jie Chen**
MIT-IBM Watson AI Lab, IBM Research
`chenjie@us.ibm.com`

## Abstract

Preconditioning is at the heart of iterative solutions of large, sparse linear systems of equations in scientific disciplines. Several algebraic approaches, which access no information beyond the matrix itself, are widely studied and used, but ill-conditioned matrices remain very challenging. We take a machine learning approach and propose using graph neural networks as a general-purpose preconditioner. They show attractive performance for many problems and can be used when the mainstream preconditioners perform poorly. Empirical evaluation on over 800 matrices suggests that the construction time of these graph neural preconditioners (GNPs) is more predictable and can be much shorter than that of other widely used ones, such as ILU and AMG, while the execution time is faster than using a Krylov method as the preconditioner, such as in inner-outer GMRES. GNPs have a strong potential for solving large-scale, challenging algebraic problems arising from not only partial differential equations, but also economics, statistics, graph, and optimization, to name a few.

## 1 Introduction

Iterative methods are commonly used to solve large, sparse linear systems of equations $\mathbf{Ax} = \mathbf{b}$. These methods typically build a Krylov subspace, onto which the original system is projected, such that an approximate solution within the subspace is extracted. For example, GMRES (Saad & Schultz, 1986), one of the most popularly used methods in practice, builds an orthonormal basis $\{\mathbf{v}_1, \mathbf{v}_2, \ldots, \mathbf{v}_m\}$ of the $m$-dimensional Krylov subspace by using the Arnoldi process and defines the approximate solution $\mathbf{x}_m \in \mathbf{x}_0 + \text{span}(\{\mathbf{v}_1, \mathbf{v}_2, \ldots, \mathbf{v}_m\})$, such that the residual norm $\|\mathbf{b} - \mathbf{Ax}_m\|_2$ is minimized.

The effectiveness of Krylov methods heavily depends on the conditioning of the matrix $\mathbf{A}$. Hence, designing a good preconditioner is crucial in practice. In some cases (e.g., solving partial differential equations, PDEs), the problem structure provides additional information that aids the development of a preconditioner applicable to this problem or a class of similar problems. For example, multigrid preconditioners are particularly effective for Poisson-like problems with a mesh. In other cases, little information is known beyond the matrix itself. An example is the SuiteSparse matrix collection (Davis & Hu, 2011), which contains thousands of sparse matrices for benchmarking numerical linear algebra algorithms. In these cases, a general-purpose (also called "algebraic") preconditioner is desirable; examples include ILU (Saad, 1994), approximate inverse (Chow & Saad, 1998), and algebraic multigrid (AMG) (Ruge & Stüben, 1987). However, it remains very challenging to design one that performs well universally.

In this work, we propose to use neural networks as a general-purpose preconditioner. Specifically, we consider preconditioned Krylov methods for solving problems in the form $\mathbf{AMu} = \mathbf{b}$, where $\mathbf{M} \approx \mathbf{A}^{-1}$ is the preconditioner, $\mathbf{u}$ is the new unknown, and $\mathbf{x} = \mathbf{Mu}$ is the recovered solution. We design a graph neural network (GNN) (Zhou et al., 2020; Wu et al., 2021) to assume the role of $\mathbf{M}$ and develop a training method to learn $\mathbf{M}$ from select $(\mathbf{b}, \mathbf{x})$ pairs. This proposal is inspired by the universal approximation property of neural networks (Hornik et al., 1989) and encouraged by their widespread success in artificial intelligence (Bommasani et al., 2021).

Because a neural network is, in nature, nonlinear, our preconditioner $\mathbf{M}$ is not a linear operator anymore. We can no longer build a Krylov subspace for $\mathbf{AM}$ when a Krylov subspace is defined for

only linear operators. Hence, we focus on flexible variants of the preconditioned Krylov methods instead. Specifically, we use flexible GMRES (FGMRES) (Saad, 1993), which considers $\mathbf{M}$ to be different in every Arnoldi step. In this framework, all what matters is the subspace from which the approximate solution is extracted. A nonlinear operator $\mathbf{M}$ can also be used to build this subspace and hence the neural preconditioner is applicable to FGMRES.

We use a *graph* neural network because of the natural connection between a sparse matrix and the graph adjacency matrix, similar to how AMG interprets the coefficient matrix with a graph. Many GNN architectures access the graph structure through $\mathbf{A}$-multiplications and they become polynomials of $\mathbf{A}$ when the nonlinearity is omitted (Wu et al., 2019; Chen et al., 2020). This observation bridges the connection between a GNN as an operator and the polynomial approximation of $\mathbf{A}^{-1}$, the latter of which is an essential ingredient in the convergence theory of Krylov methods. Interestingly, a side benefit resulting from the access pattern of $\mathbf{A}$ is that the GNN is applicable to not only sparse matrices, but also structured matrices that admit fast $\mathbf{A}$-multiplications (such as hierarchical matrices (Hackbusch, 1999; Hackbusch & Börm, 2002; Chen & Stein, 2023), Toeplitz matrices (Chan & Jin, 2007; Chen et al., 2014), unassembled finite-element matrices (Ciarlet, 2002), and Gauss-Newton matrices (Liu et al., 2023)).

There are a few advantages of the proposed graph neural preconditioner (GNP). It performs comparatively much better for ill-conditioned problems, by learning the matrix inverse from data, mitigating the restrictive modeling of the sparsity pattern (as in ILU and approximate inverse), the insufficient quality of polynomial approximation (as in using Krylov methods as the preconditioner), and the challenge of smoothing over a non-geometric mesh (as in AMG). Additionally, empirical evaluation suggests that the construction time of GNP is more predictable, inherited from the predictability of neural network training, than that of ILU and AMG; and the execution time of GNP is much shorter than GMRES as the preconditioner, which may be bottlenecked by the orthogonalization in the Arnoldi process.

## 1.1 CONTRIBUTIONS

Our work has a few technical contributions. First, we offer a convergence analysis for FGMRES. Although this method is well known and used in practice, little theoretical work was done, in part because the tools for analyzing Krylov subspaces are not readily applicable (as the subspace is no longer Krylov and we have no isomorphism with the space of polynomials). Instead, our analysis is a posteriori, based on the computed subspace.

Second, we propose an effective approach to training the neural preconditioner; in particular, training data generation. While it is straightforward to train the neural network in an online-learning manner—through preparing streaming, random input-output pairs in the form of $(\mathbf{b}, \mathbf{x})$—it leaves open the question regarding what randomness best suits the purpose. We consider the bottom eigensubspace of $\mathbf{A}$ when defining the sampling distribution and show the effectiveness of this approach.

Third, we develop a scale-equivariant GNN as the preconditioner. Because of the way that training data are generated, the GNN inputs $\mathbf{b}$ can have vast different scales; meanwhile, the ground truth operator $\mathbf{A}^{-1}$ is equivariant to the scaling of the inputs. Hence, to facilitate training, we design the GNN by imposing an inductive bias that obeys scale-equivariance.

Fourth, we adopt a novel evaluation protocol for general-purpose preconditioners. The common practice evaluates a preconditioner on a limited number of problems; in many occasions, these problems come from a certain type of PDEs or an application, which constitutes only a portion of the use cases. To broadly evaluate a new kind of general-purpose preconditioner and understand its strengths and weaknesses, we propose to test it on a large number of matrices commonly used by the community. To this end, we use the SuiteSparse collection and perform evaluation on a substantial portion of it (*all* non-SPD, real matrices falling inside a size interval; over 800 from 50 application areas in this work). To streamline the evaluation, we define evaluation criteria, limit the tuning of hyperparameters, and study statistics over the distribution of matrices.

## 1.2 RELATED WORK

It is important to put the proposed GNP in context. The idea of using a neural network to construct a preconditioner emerged recently. The relevant approaches learn the nonzero entries of the

incomplete factors (Li et al., 2023; Häusner et al., 2023) or their correction (Trifonov et al., 2024), or of the approximate inverse (Bånkestad et al., 2024). In these approaches, the neural network does not directly approximate the mapping from $\mathbf{b}$ to $\mathbf{x}$ like ours does; rather, it is used to complete the nonzeros of the predefined sparsity structure. A drawback of these approaches is that the predefined sparsity imposes a nonzero lower bound on the approximation of $\mathbf{A}^{-1}$, which (and whose factors) are generally not sparse. Moreover, although the neural network can be trained on a PDE problem (by varying the coefficients, grid sizes, or boundary conditions) and generalizes well to the same problem, it is unlikely that a single network can work well for all problems and matrices in life.

Approaches wherein the neural network directly approximates the matrix inverse are more akin to physics-informed neural networks (PINNs) (Raissi et al., 2019) and neural operators (NOs) (Rudikov et al., 2024; Kovachki et al., 2022; Li et al., 2020; 2021; Lu et al., 2021). However, the learning of PINNs requires a PDE whose residual forms a partial training loss, whereas NOs consider infinite-dimensional function spaces, with PDEs being the application. In our case of a general-purpose preconditioner, there is not an underlying PDE; and not every matrix problem relates to function spaces with additional properties to exploit (e.g., spatial coordinates, smoothness, and decay of the Green's function). For example, one may be interested in finding the commute times between a node and all other nodes in a graph; this problem can be solved by solving a linear system with respect the graph Laplacian matrix. The only information we exploit is the matrix itself.

## 2 METHOD

Let $\mathbf{A} \in \mathbb{R}^{n \times n}$. From now on, $\mathbf{M}$ is an operator $\mathbb{R}^n \to \mathbb{R}^n$ and we write $\mathbf{M}(\mathbf{v})$ to mean applying the operator on $\mathbf{v} \in \mathbb{R}^n$. This notation includes the special case $\mathbf{M}(\mathbf{v}) = \mathbf{M}\mathbf{v}$ when $\mathbf{M}$ is a matrix.

### 2.1 FLEXIBLE GMRES

A standard preconditioned Krylov solver solves the linear system $\mathbf{A}\mathbf{M}\mathbf{u} = \mathbf{b}$ by viewing $\mathbf{A}\mathbf{M}$ as the new matrix and building a Krylov subspace, from which an approximate solution $\mathbf{u}_m$ is extracted and $\mathbf{x}_m$ is recovered from $\mathbf{M}\mathbf{u}_m$. It is important to note that a subspace, by definition, is *linear*. However, applying a nonlinear operator $\mathbf{M}$ on the linear vector space does not result in a vector space spanned by the mapped basis. Hence, convergence theory of $\mathbf{x}_m = \mathbf{M}(\mathbf{u}_m)$ is broken.

We resort to flexible variants of Krylov solvers (Saad, 1993; Notay, 2000; Chen et al., 2016) and focus on flexible GMRES for simplicity. FGMRES was designed such that the matrix $\mathbf{M}$ can change in each Arnoldi step; we extend it to a fixed, but nonlinear, operator $\mathbf{M}$.

Algorithm 1 in Section B summarizes the solver. The algorithm assumes a restart length $m$. Starting from an initial guess $\mathbf{x}_0$ with residual vector $\mathbf{r}_0$ and 2-norm $\beta$, FGMRES runs $m$ Arnoldi steps, resulting in the relation

$$\mathbf{A}\mathbf{Z}_m = \mathbf{V}_m\mathbf{H}_m + h_{m+1,m}\mathbf{v}_{m+1}\mathbf{e}_m^\top = \mathbf{V}_{m+1}\overline{\mathbf{H}}_m, \tag{1}$$

where $\mathbf{V}_m = [\mathbf{v}_1, \ldots, \mathbf{v}_m] \in \mathbb{R}^{n \times m}$ is orthonormal, $\mathbf{Z}_m = [\mathbf{z}_1, \ldots, \mathbf{z}_m] \in \mathbb{R}^{n \times n}$ contains $m$ columns $\mathbf{z}_j = \mathbf{M}(\mathbf{v}_j)$, $\mathbf{H}_m = [h_{ij}] \in \mathbb{R}^{m \times m}$ is upper Hessenberg, and $\overline{\mathbf{H}}_m = [h_{ij}] \in \mathbb{R}^{(m+1) \times m}$ extends $\mathbf{H}_m$ with an additional row at the bottom. The approximate solution $\mathbf{x}_m$ is computed in the form $\mathbf{x}_m \in \mathbf{x}_0 + \text{span}(\{\mathbf{z}_1, \ldots, \mathbf{z}_m\}) = \{\mathbf{x}_0 + \mathbf{Z}_m\mathbf{y}\}$, such that the residual norm $\|\mathbf{b} - \mathbf{A}\mathbf{x}_m\|_2 = \|\beta\mathbf{e}_1 - \overline{\mathbf{H}}_m\mathbf{y}\|_2$ is minimized. If the residual norm is sufficiently small, the solution is claimed; otherwise, let $\mathbf{x}_m$ be the initial guess of the next Arnoldi cycle and repeat.

We provide a convergence analysis below. All proofs are in Section C.

**Theorem 1.** *Assume that FGMRES is run without restart and without breakdown. We use the tilde notation to denote the counterpart quantities when FGMRES is run with a fixed preconditioner matrix $\widetilde{\mathbf{M}}$. For any $\widetilde{\mathbf{M}}$ such that $\mathbf{A}\widetilde{\mathbf{M}}$ can be diagonalized, as in $\mathbf{X}^{-1}\mathbf{A}\widetilde{\mathbf{M}}\mathbf{X} = \mathbf{\Lambda} = \text{diag}(\lambda_1, \ldots, \lambda_n)$, the residual $\mathbf{r}_m = \mathbf{b} - \mathbf{A}\mathbf{x}_m$ satisfies*

$$\|\mathbf{r}_m\|_2 \leq \kappa_2(\mathbf{X})\epsilon^{(m)}(\mathbf{\Lambda})\|\mathbf{r}_0\|_2 + \|\mathbf{Q}_m\mathbf{Q}_m^\top - \widetilde{\mathbf{Q}}_m\widetilde{\mathbf{Q}}_m^\top\|_2\|\mathbf{r}_0\|_2, \tag{2}$$

*where $\kappa_2$ denotes the 2-norm condition number, $\mathbf{Q}_m$ (resp. $\widetilde{\mathbf{Q}}_m$) is the thin-Q factor of $\mathbf{A}\mathbf{Z}_m$ (resp. $\mathbf{A}\widetilde{\mathbf{Z}}_m$), $\mathbb{P}_m$ denotes the space of degree-$m$ polynomials, and*

$$\epsilon^{(m)}(\mathbf{\Lambda}) = \min_{p \in \mathbb{P}_m, \, p(0)=1} \max_{i=1,\ldots,n} |p(\lambda_i)|.$$

Let us consider the two terms on the right-hand side of (2). The first term $\kappa_2(\mathbf{X})\epsilon^{(m)}(\mathbf{\Lambda})\|\mathbf{r}_0\|_2$ is the standard convergence result of GMRES on the matrix $\mathbf{A}\widetilde{\mathbf{M}}$. The factor $\epsilon^{(m)}(\mathbf{\Lambda})$ is approximately exponential in $m$, when the eigenvalues $\lambda_i$ are located in an ellipse which excludes the origin; see Saad (2003, Corollary 6.33). Furthermore, when the eigenvalues are closer to each other than to the origin, the base of the exponential is close to zero, resulting in rapid convergence. Meanwhile, when $\mathbf{A}\widetilde{\mathbf{M}}$ is close to normal, $\kappa_2(\mathbf{X})$ is close to 1.

What $\mathbf{M}$ affects is the second term $\|\mathbf{Q}_m\mathbf{Q}_m^\top - \widetilde{\mathbf{Q}}_m\widetilde{\mathbf{Q}}_m^\top\|_2\|\mathbf{r}_0\|_2$. This term does not show convergence on the surface, but one can always find an $\widetilde{\mathbf{M}}$ such that it vanishes, assuming no breakdown. When $\mathbf{M}$ is close to $\mathbf{A}^{-1}$, $\mathbf{A}\mathbf{Z}_m$ is a small perturbation of $\mathbf{V}_m$ by definition. Then, (1) suggests that $\overline{\mathbf{H}}_m$ is close to the identity matrix for all $m$ and so is $\mathbf{H}_n$.

**Corollary 2.** *Assume that FGMRES is run without restart and without breakdown. On completion, let $\mathbf{H}_n$ be diagonalizable, as in $\mathbf{Y}^{-1}\mathbf{H}_n\mathbf{Y} = \mathbf{\Sigma} = \mathrm{diag}(\sigma_1,\ldots,\sigma_n)$. Then, the residual norm satisfies*

$$\|\mathbf{r}_m\|_2 \leq \kappa_2(\mathbf{Y})\epsilon^{(m)}(\mathbf{\Sigma})\|\mathbf{r}_0\|_2 \qquad \textit{for all } m. \tag{3}$$

See Section D for an approximately exponential convergence of $\|\mathbf{r}_m\|_2$ due to $\epsilon^{(m)}(\mathbf{\Sigma})$.

Note that while our results assume infinite precision, another analysis (Arioli & Duff, 2009) was conducted based on finite arithmetics. Such analysis uses the machine precision to bound the residual norm for a sufficiently large $m$, asserting backward stability of FGMRES; on the other hand, we can obtain the convergence rate under infinite precision.

## 2.2 Training Data Generation

We generate a set of $(\mathbf{b}, \mathbf{x})$ pairs to train $\mathbf{M} : \mathbf{b} \rightarrow \mathbf{x}$. Unlike neural operators where the data generation is costly (e.g., requiring solving PDEs), which limits the training set size, for linear systems, we can sample $\mathbf{x}$ from a certain distribution and obtain $\mathbf{b} = \mathbf{A}\mathbf{x}$ with negligible cost, which creates a streaming training set of $(\mathbf{b}, \mathbf{x})$ pairs without size limit.[1]

There is no free lunch. One may sample $\mathbf{x} \sim \mathcal{N}(\mathbf{0}, \mathbf{I}_n)$, which leads to $\mathbf{b} \sim \mathcal{N}(\mathbf{0}, \mathbf{A}\mathbf{A}^\top)$. This distribution is skewed toward the dominant eigen-subspace of $\mathbf{A}\mathbf{A}^\top$. Hence, using samples from it for training may result in a poor performance of the preconditioner, when applied to inputs lying close to the bottom eigen-subspace. One may alternatively want the training data $\mathbf{b} \sim \mathcal{N}(\mathbf{0}, \mathbf{I}_n)$, which covers uniformly all possibilities of the preconditioner input on a sphere. However, in this case, $\mathbf{x} \sim \mathcal{N}(\mathbf{0}, \mathbf{A}^{-1}\mathbf{A}^{-\top})$ is also skewed in the spectrum, causing difficulty in network training.

We resort to the Arnoldi process to obtain an approximation of $\mathbf{A}^{-1}\mathbf{A}^{-\top}$, which strikes a balance. Specifically, we run the Arnoldi process in $m$ steps without a preconditioner, which results in a simplification of (1):

$$\mathbf{A}\mathbf{V}_m = \mathbf{V}_m\mathbf{H}_m + h_{m+1,m}\mathbf{v}_{m+1}\mathbf{e}_m^\top = \mathbf{V}_{m+1}\overline{\mathbf{H}}_m. \tag{4}$$

Let the singular value decomposition of $\overline{\mathbf{H}}_m$ be $\mathbf{W}_m\mathbf{S}_m\mathbf{Z}_m^\top$, where $\mathbf{W}_m \in \mathbb{R}^{(m+1)\times m}$ and $\mathbf{Z}_m \in \mathbb{R}^{m\times m}$ are orthonormal and $\mathbf{S}_m \in \mathbb{R}^{m\times m}$ is diagonal. We define

$$\mathbf{x} = \mathbf{V}_m\mathbf{Z}_m\mathbf{S}_m^{-1}\boldsymbol{\epsilon}, \qquad \boldsymbol{\epsilon} \sim \mathcal{N}(\mathbf{0}, \mathbf{I}_m). \tag{5}$$

Then, with simple algebra we obtain that

$$\mathbf{x} \in \mathrm{range}(\mathbf{V}_m), \qquad \mathbf{x} \sim \mathcal{N}(\mathbf{0}, \mathbf{\Sigma}_m^{\mathbf{x}}), \qquad \mathbf{\Sigma}_m^{\mathbf{x}} = (\mathbf{V}_m\overline{\mathbf{H}}_m^+)(\mathbf{V}_m\overline{\mathbf{H}}_m^+)^\top,$$

$$\mathbf{b} \in \mathrm{range}(\mathbf{V}_{m+1}), \qquad \mathbf{b} \sim \mathcal{N}(\mathbf{0}, \mathbf{\Sigma}_m^{\mathbf{b}}), \qquad \mathbf{\Sigma}_m^{\mathbf{b}} = (\mathbf{V}_{m+1}\mathbf{W}_m)(\mathbf{V}_{m+1}\mathbf{W}_m)^\top.$$

The covariance of $\mathbf{b}$, $\mathbf{\Sigma}_m^{\mathbf{b}}$, is a projector, because $\mathbf{V}_{m+1}$ and $\mathbf{W}_m$ are orthonormal. As $m$ grows, this covariance tends to the identity $\mathbf{I}_n$. By the theory of the Arnoldi process (Morgan, 2002), $\mathbf{V}_{m+1}$ contains better and better approximations of the extreme eigenvectors as $m$ increases. Hence, $\mathcal{N}(\mathbf{0}, \mathbf{\Sigma}_m^{\mathbf{b}})$ has a high density close to the bottom eigen-subspace of $\mathbf{A}$. Meanwhile, because $\mathbf{\Sigma}_m^{\mathbf{b}}$ is

---

[1]Neural operators learn the mapping from the boundary condition to the solution. Strictly speaking, while $\mathbf{x}$ is the solution, $\mathbf{b}$ is not the boundary condition.

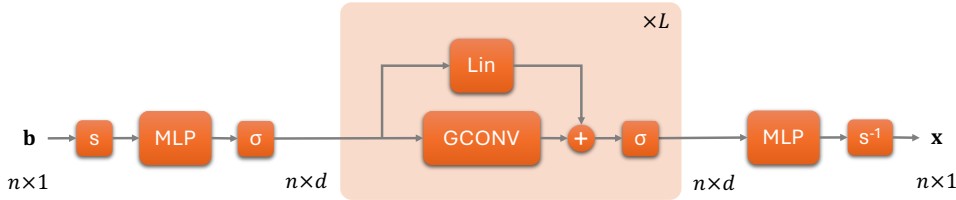

Figure 1: Our GNN architecture for the preconditioner $\mathbf{M}$. The nonlinearity $\sigma$ is ReLU. The component $s$ denotes the scale-equivariant operator (8).

low-rank, the distribution has zero density along many directions. Therefore, we sample $\mathbf{x}$ from both $\mathcal{N}(\mathbf{0}, \boldsymbol{\Sigma}_m^{\mathbf{x}})$ and $\mathcal{N}(\mathbf{0}, \mathbf{I}_n)$ to form each training batch, to cover more important subspace directions.

We note that the use of Gaussian distributions brings in the convenience to generally analyze the input/output distributions. It is possible that other distributions are more effective if knowledge of $\mathbf{x}$ exists and can be exploited.

### 2.3 SCALE-EQUIVARIANT GRAPH NEURAL PRECONDITIONER

We now define the neural network $\mathbf{M}$ that approximates $\mathbf{A}^{-1}$. Naturally, a (sparse) matrix $\mathbf{A} = [a_{ij}]$ admits a graph interpretation, similar to how AMG interprets the coefficient matrix with a graph. Treating $\mathbf{A}$ as the graph adjacency matrix allows us to use GNNs (Kipf & Welling, 2017; Hamilton et al., 2017; Veličković et al., 2018; Xu et al., 2019) to parameterize the preconditioner. These neural networks perform graph convolutions in a neighborhood of each node, reducing the quadratic pairwise cost by ignoring nodes faraway from the neighborhood.

Our neural network is based on the graph convolutional network (GCN) (Kipf & Welling, 2017). A GCN layer is defined as $\mathrm{GCONV}(\mathbf{X}) = \mathrm{ReLU}(\widehat{\mathbf{A}}\mathbf{X}\mathbf{W})$, where $\widehat{\mathbf{A}} \in \mathbb{R}^{n \times n}$ is some normalization of $\mathbf{A}$, $\mathbf{X} \in \mathbb{R}^{n \times d_{\mathrm{in}}}$ is the input data, and $\mathbf{W} \in \mathbb{R}^{d_{\mathrm{in}} \times d_{\mathrm{out}}}$ is a learnable parameter.

Our neural network, as illustrated in Figure 1, enhances from GCN in a few manners:

1. We normalize $\mathbf{A}$ differently. Specifically, we define

$$\widehat{\mathbf{A}} = \mathbf{A}/\gamma \quad \text{where} \quad \gamma = \min\left\{ \max_i \left\{ \sum_j |a|_{ij} \right\}, \max_j \left\{ \sum_i |a|_{ij} \right\} \right\}. \tag{6}$$

   The factor $\gamma$ is an upper bound of the spectral radius of $\mathbf{A}$ based on the Gershgorin circle theorem (Gerschgorin, 1931). This avoids division-by-zero that may occur in the standard GCN normalization, because the nonzeros of $\mathbf{A}$ can be both positive and negative.

2. We add a residual connection to GCONV, to allow stacking a deeper GNN without suffering the smoothing problem (Chen et al., 2020). Specifically, we define a Res-GCONV layer to be

$$\mathrm{Res\text{-}GCONV}(\mathbf{X}) = \mathrm{ReLU}(\mathbf{X}\mathbf{U} + \widehat{\mathbf{A}}\mathbf{X}\mathbf{W}). \tag{7}$$

   Moreover, we let the parameters $\mathbf{U}$ and $\mathbf{W}$ be square matrices of dimension $d \times d$, such that the input/output dimensions across layers do not change. We stack $L$ such layers.

3. We use an MLP encoder (resp. MLP decoder) before (resp. after) the $L$ Res-GCONV layers to lift (resp. project) the dimensions.

Additionally, the main novelty of the architecture is scale-equivariance. Note that the operator to approximate, $\mathbf{A}^{-1}$, is scale-equivariant, but a general neural network is not guaranteed so. Because the input $\mathbf{b}$ may have very different scales (due to the sampling approach considered in the preceding subsection), to facilitate training, we design a parameter-free scaling $s$ and back-scaling $s^{-1}$:

$$s(\cdot) = \frac{\sqrt{n}}{\tau}\cdot \quad \text{and} \quad s^{-1}(\cdot) = \frac{\tau}{\sqrt{n}}\cdot, \quad \text{where } \tau = \|\mathbf{b}\|_2. \tag{8}$$

We apply $s$ at the beginning and $s^{-1}$ at the end of the neural network. Effectively, we restrict the input space of the neural network from the full $\mathbb{R}^n$ to a more controllable space—the sphere $\sqrt{n}\mathbb{S}^{n-1}$. Clearly, $s$ guarantees scale-equivariance; i.e., $\mathbf{M}(\alpha\mathbf{b}) = \alpha\mathbf{M}(\mathbf{b})$ for any scalar $\alpha \neq 0$.

## 3  EVALUATION METHODOLOGY AND EXPERIMENT SETTING

**Problems.** Our evaluation strives to cover a large number of problems, which come from as diverse application areas as possible. To this end, we turn to the SuiteSparse matrix collection `https://sparse.tamu.edu`, which is a widely used benchmark in numerical linear algebra. We select *all* square, real-valued, and non-SPD matrices whose number of rows falls between 1K and 100K and whose number of nonzeros is fewer than 2M. This selection results in 867 matrices from 50 application areas. Some applications are involved with PDEs (such as computational fluid dynamics), while others come from graph, optimization, economics, and statistics problems.

To facilitate solving the linear systems, we prescale each $\mathbf{A}$ by $\gamma$ defined in (6). This scaling is a form of normalization, which avoids the overall entries of the matrix being exceedingly small or large. All experiments assume the ground truth solution $\mathbf{x} = \mathbf{1}$.[2] FGMRES starts with $\mathbf{x}_0 = \mathbf{0}$.

**Compared methods.** We compare GNP with three widely used general-purpose preconditioners—ILU, AMG, and GMRES (using GMRES to precondition GMRES is also called inner-outer GMRES)—and one simple-to-implement but weak preconditioner—Jacobi. The endeavor of preconditioning over 800 matrices from different domains limits the choices, but these preconditioners are handy with Python software support.

ILU comes from `scipy.sparse.linalg.spilu`; it in turn comes from SuperLU (Li, 2005; Li & Shao, 2010), which implements the thresholded version of ILU (Saad, 1994). We use the default drop tolerance and fill factor without tuning. AMG comes from PyAMG (Bell et al., 2023). Specifically, we use `pyamg.blackbox.solver().aspreconditioner` as the preconditioner, with the configuration of the blackbox solver computed from `pyamg.blackbox.solver_configuration`. GMRES is self-implemented. For (the inner) GMRES, we use 10 iterations and stop it when it reaches a relative residual norm tolerance of `1e-6`. For (the outer) FGMRES, the restart cycle is 10.

Note that sophisticated preconditioners require tremendous efforts to implement robustly. Hence, we prioritize ones that are more robust and that come with continual supports. For example, we also experiment with AmgX (Naumov et al., 2015) but find that while being faster, it throws more errors than does PyAMG. See Section F.4 for details.

**Stopping criteria.** The stopping criteria of all linear system solutions are `rtol = 1e-8` and `maxiters = 100`. For time comparisons, we additionally run experiments by enabling `timeout` and disabling `maxiters`. We set the timeout to be the maximum solution time among all preconditioners when using `maxiters = 100` as the stopping criterion.

**Evaluation metrics.** Evaluating the solution qualities and comparing different methods for a large number of problems require programmable metrics, but defining these metrics is harder than it appears to be. Comparing the number of iterations alone is insufficient, because the per-iteration time of each method is different. However, to compare time, one waits until the solution reaches `rtol`, which is impossible to set if one wants *all* solutions to reach this tolerance. Thus, we can use `maxiters` and `timeout` to terminate a solution. However, one solution may reach `maxiters` early (in time) and attain poor accuracy, while another solution may reach `timeout` very late but attain good accuracy. Which one is better is debatable.

Hence, we propose two novel metrics, depending on the use of the stopping criteria. The first one, *area under the relative residual norm curve with respect to iterations*, is defined as

$$\text{Iter-AUC} = \sum_{i=0}^{\text{iters}} \log_{10} r_i - \log_{10} \texttt{rtol}, \quad r_i = \|\mathbf{b} - \mathbf{A}\mathbf{x}_i\|_2 / \|\mathbf{b}\|_2,$$

where `iters` is the actual number of iterations when FGMRES stops and $r_i$ is the relative residual norm at iteration $i$. We take logarithm because residual plots are typically done in the log scale. This metric compares methods based on the iteration count, taking into account the history (e.g., convergence speed at different stages) rather than the final error alone. This metric is used when the stopping criteria are `rtol` and `maxiters`.

---

[2]This is a common practice. Avoid setting $\mathbf{x}$ to be Gaussian random, because this (partially) matches the training data generation, resulting in the inability to test out-of-distribution cases.

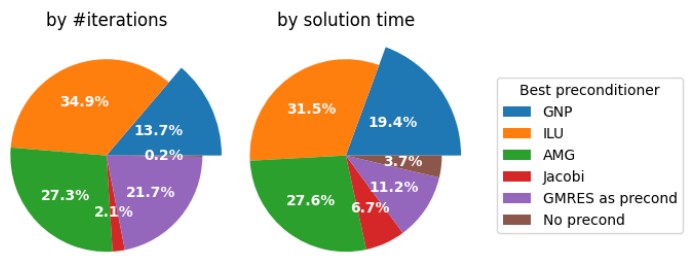

Figure 2: Percentage of problems on which each preconditioner performs the best.

Table 1: Distribution of the residual-norm ratio between the second best preconditioner and GNP, when GNP performs the best. Distribution is described by using percentiles.

|  | by #iter | by time |
|---|---|---|
| 25% | 1.91e+00 | 1.18e+00 |
| 50% | 6.74e+00 | 2.33e+00 |
| 75% | 6.78e+03 | 8.16e+01 |
| 100% | 1.89e+10 | 1.91e+10 |

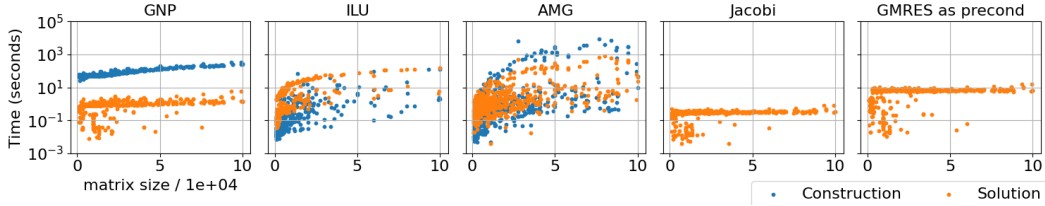

Figure 3: Preconditioner construction time and solution time (using `maxiters` to stop). The construction time of Jacobi is negligible and not shown. GMRES does not require construction.

The second metric, *area under the relative residual norm curve with respect to time*, is defined as

$$\text{Time-AUC} = \int_0^T \left[\log_{10} r(t) - \log_{10} \texttt{rtol}\right] dt \approx \sum_{i=1}^{\text{iters}} \left[\log_{10} r_i - \log_{10} \texttt{rtol}\right](t_i - t_{i-1}),$$

where $T$ is the elapsed time when FGMRES stops, $r(t)$ is the relative residual norm at time $t$, and $t_i$ is the elapsed time at iteration $i$. This metric compares methods based on the solution time, taking into account the history of the errors. It is used when the stopping criteria are `rtol` and `timeout`.

**Hyperparameters.** To feasibly train over 800 GNNs, we use the same set of hyperparameters for each of them without tuning. There is a strong potential that the GNN performance can be greatly improved with careful tuning. Our purpose is to understand the general behavior of GNP, particularly its strengths and weaknesses. We use $L = 8$ Res-GCONV layers, set the layer input/output dimension to 16, and use 2-layer MLPs with hidden dimension 32 for lifting/projection. We use Adam (Kingma & Ba, 2015) as the optimizer, set the learning rate to `1e-3`, and train for 2000 steps with a batch size of 16. We apply neither dropouts nor weight decays. Because data are sampled from the same distribution, we use the model at the best training epoch as the preconditioner, without resorting to a validation set.

We use the $\ell_1$ residual norm $\|\mathbf{A}\mathbf{M}(\mathbf{b}) - \mathbf{A}\mathbf{x}\|_1$ as the training loss, as a common practice of robust regression. We use $m = 40$ Arnoldi steps when sampling the $(\mathbf{x}, \mathbf{b})$ pairs according to (5). Among the 16 pairs in a batch, 8 pairs follow (5) and 8 pairs follow $\mathbf{x} \sim \mathcal{N}(\mathbf{0}, \mathbf{I}_n)$.

**Compute environment.** Our experiments are conducted on a machine with one Tesla V100(16GB) GPU, 96 Intel Xeon 2.40GHz cores, and 386GB main memory. All code is implemented in Python with Pytorch. All computations (training and inference) use the GPU whenever supported.

## 4 RESULTS

We organize the experiment results by key questions of interest.

**How does GNP perform compared with traditional preconditioners?** We consider three factors: convergence speed, runtime speed, and preconditioner construction cost.

In Figure 2 we show the percentage of problems on which each preconditioner performs the best, with respect to iteration counts and solution time (see the Iter-AUC and Time-AUC metrics defined

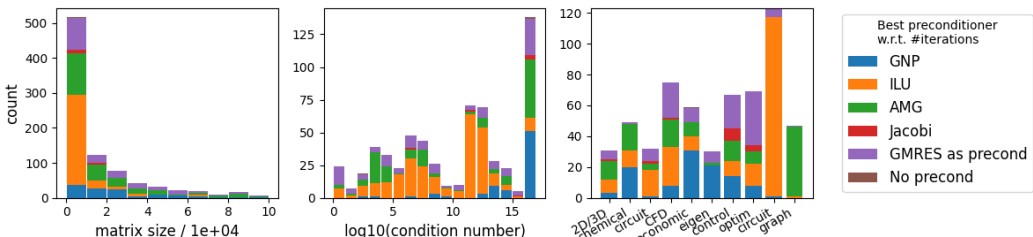

Figure 4: Breakdown of best preconditioners with respect to matrix sizes, condition numbers, and application areas. Only the application areas with the top number of problems are shown. The last bar in the middle plot is for condition number $\geq 10^{16}$.

in the preceding section). GNP performs the best for a substantial portion of the problems in both metrics. In comparison, ILU and AMG perform the best for more problems, while Jacobi, GMRES (as a preconditioner), and no preconditioner performs the best for fewer problems in the time metric.

That ILU and AMG are the best for more problems does not compromise the competitiveness of GNP. For one reason, they are less robust and sometimes bare a significantly higher cost in construction, as will be revealed later. Another reason is that GNP can be used when other preconditioners are less satisfactory. In Table 1, we summarize the distribution of the residual-norm ratio between the second best preconditioner and GNP, when GNP is the best. The ratio means a factor of $x$ decrease in the residual norm if GNP is used in place of other preconditioners. Under the Iter-AUC metric, a factor of 6780x decrease can be seen at the 75th percentile, and in the best occasion, more than ten orders of magnitude decrease is seen. Similarly under the Time-AUC metric. These significant reductions manifest the usefulness of GNP.

In Figure 3 and Figure 8 of Section F.2, we plot the preconditioner construction time and solution time for each matrix and each preconditioner. Here, the solution is terminated by `maxiters` rather than `timeout` (such that the times are different across matrices). Two observations can be made. First, the training time of GNP is nearly proportional to the matrix size, while the construction time of ILU and AMG is hardly predictable. Even though for many of the problems, the time is only a fraction of that of GNP, there exist quite a few cases where the time is significantly longer. In the worst case, the construction of the AMG preconditioner is more than an order of magnitude more costly than that of GNP.

Second, there is a clear gap between the construction and solution time for GNP, but no such gap exists for ILU and AMG. One is tempted to compare the overall time, but one preconditioner can be used for any number of right-hand sides, and hence different conclusions may be reached regarding time depending on this number. When it is large, the solution time dominates, in which case one sees that in most of the cases GNP is faster ILU, AMG, and GMRES.

**On what problems does GNP perform the best?** We expand the granularity of Figure 2 into Figure 4, overlaid with the distributions of the matrices regarding the size, the condition number, and the application area, respectively. One finding is that GNP is particularly useful for ill-conditioned matrices (i.e., those with a condition number $\geq 10^{16}$). ILU is less competitive in these problems, possibly because it focuses on the sparsity structure, whose connection with the spectrum is less clear. Another finding is that GNP is particularly useful in three application areas: chemical process simulation problems, economic problems, and eigenvalue/model reduction problems. These areas constitute many problems in the dataset; both the number and the proportion of them in which GNP performs the best are substantially high.

On the other hand, the matrix size does not affect much the GNP performance. One sees this more clearly when consulting Figure 9 in Section F.3, which is a "proportion" version of Figure 4: the proportion of problems on which GNP performs the best is relatively flat across matrix sizes. Similarly, symmetry does not appear to play a distinguishing role either: GNP performs the best on 7.6% of the symmetric matrices and 17.5% of the nonsymmetric matrices.

**How robust is GNP?** We summarize in Table 2 the number of failures for each preconditioner. GNP and GMRES are highly robust. We see that neural network training does not cause any troubles, which is a practical advantage. In contrast, ILU fails for nearly half of the problems, AMG fails for

Table 2: Failures of preconditioners (count and proportion).

| | GNP | ILU | AMG | Jacobi | GMRES as precond |
|---|---|---|---|---|---|
| Construction failure | 0 (0.00%) | 348 (40.14%) | 62 (7.15%) | N/A | N/A |
| Solution failure | 1 (0.12%) | 61 ( 7.04%) | 5 (0.58%) | 53 (6.11%) | 2 (0.23%) |

Figure 5: Example convergence histories.

nearly 8%, and Jacobi fails for over 6%. According to the error log, the common failures of ILU are that "(f)actor is exactly singular" and that "matrix is singular ... in file ilu_dpivotL.c", while the common failures of AMG are "array ... contain(s) infs or NaNs". Meanwhile, solution failures occur when the residual norm tracked by the QR factorization of the upper Hessenberg $\overline{\mathbf{H}}_m$ fails to match the actual residual norm. While ILU and AMG perform the best for more problems than does GNP, safeguarding their robustness is challenging, rendering GNP more attractive.

**What does the convergence history look like?** In Figure 5, we show a few examples when GNP performs the best. Note that in this case, ILU fails for most of the problems, in either the construction or the solution phase. The examples indicate that the convergence of GNP either tracks that of other preconditioners with a similar rate (`Simon/venkat25`), or becomes significantly faster. Notably, on `VanVelzen/std1_Jac3`, GNP uses only four iterations to reach `1e-8`, while other preconditioners take 100 iterations but still cannot decrease the relative residual to below `1e-2`.

We additionally plot the convergence curves with respect to time in Figure 10 in Section F.6. An important finding to note is that GMRES as a preconditioner generally takes much longer time to run than GNP. This is because GMRES is bottlenecked by the orthogonalization process, shadowing the trickier cost comparison between more matrix-vector multiplications in GMRES and fewer matrix-matrix multiplications in GNP.

**Are the proposed training-data generation and the scale-equivariance design necessary?** In Figure 6, we compare the proposed designs versus alternatives, for both the training loss and the relative residual norm. In the comparison of training data generation, using $\mathbf{x} \sim \mathcal{N}(\mathbf{0}, \mathbf{I}_n)$ leads to lower losses, while using $\mathbf{b} \sim \mathcal{N}(\mathbf{0}, \mathbf{I}_n)$ leads to higher losses. This is expected, because the former expects similar outputs from the neural network, which is easier to train, while the latter expects drastically different outputs, which make the network difficult to train. Using $\mathbf{x}$ from the proposed mixture leads to losses lying in between. This case leads to the best preconditioning performance—more problems having lower residual norms (especially those near `rtol`). In other words, the proposed training data generation strikes the best balance between easiness of neural network training and goodness of the resulting preconditioner.

In the comparison of using the scaling operator $s$ versus not, we see that the training behaviors barely differ between the two choices, but using scaling admits an advantage that more problems result in lower residual norms. These observations corroborate the choices we make in the neural network design and training data generation.

## 5 DISCUSSIONS AND CONCLUSIONS

We have presented a GNN approach to preconditioning Krylov solvers. This is the first work in our knowledge to use a neural network as an approximation to the matrix inverse, without exploiting

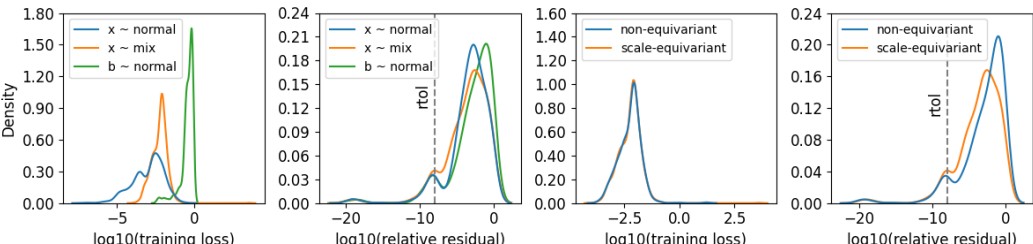

Figure 6: Left: comparison of training data generation; right: comparison of scale-equivariance.

information beyond the matrix itself (i.e., an algebraic preconditioenr). Compared with traditional algebraic preconditioners, this approach is robust with predictable construction costs and is the most effective in many problems. We evaluate the approach on more than 800 matrices from 50 application areas, the widest coverage in the literature.

One common skepticism on neural network approaches for preconditioning is that in general, network training is costly and in our case, the network can be used for only one $\mathbf{A}$. We have shown to the contrary, our training cost sometimes can be much lower than the construction cost of widely used preconditioners such as ILU and AMG, because network training is more predictable while the other preconditioners suffer the manipulation of the irregular nonzero patterns. Moreover, a benefit of neural networks is that compute infrastructures and library supports are in place, which facilitate implementation and offer robustness, as opposed to other algebraic preconditioners whose implementations are challenging and robustness is extremely difficult to achieve. Additionally, traditional algebraic preconditioners are constructed for only one $\mathbf{A}$ as well, not more flexible than ours.

While one may hope that a single neural network can solve many, if not all, linear systems (different $\mathbf{A}$'s), as recent approaches (PINN, NO, or learning the incomplete factors) appear to suggest, it is unrealistic to expect that a single network has the capacity to learn the matrix inverse for all matrices. Existing approaches work on a distribution of linear systems for the *same* problem and generalize under the same problem (e.g., by varying the grid size or PDE coefficients). For GNP to achieve so, we may fine-tune a trained network with a cost lower than training from scratch, or augment the GNN input with extra information (e.g., PDE coefficients). There unlikely exists an approach that remains effective beyond the problem being trained on.

Our work can be extended in many avenues. First, an immediate follow up is the preconditioning of SPD matrices. While a recent work on PDEs (Rudikov et al., 2024) shows promise on the combined use of NOs and flexible CG (Notay, 2000), it is relatively fragile with respect to a large variation of the preconditioner in other problems. We speculate that a split preconditioner can work more robustly and some form of autoencoder networks better serves this purpose.

Second, while GNP is trained for an individual matrix in this work for simplicity, it is possible to extend it to a sequence of evolving matrices, such as in sequential problems or time stepping. Continual learning (Wang et al., 2024), which is concerned with continuously adapting a trained neural network for slowly evolving data distributions, can amortize the initial preconditioner construction cost with more and more linear systems in sequel.

Third, the GPU memory limits the size of the matrices and the complexity of the neural networks that we can experiment with. Future work can explore multi-GPU and/or distributed training of GNNs (Kaler et al., 2022; 2023) for scaling GNP to even larger matrices. The training for large graphs typically uses neighborhood sampling to mitigate the "neighborhood explosion" problem in GNNs (Hamilton et al., 2017; Chen et al., 2018). Some sampling approaches, such as layer-wise sampling (Chen et al., 2018), are tied to sketching the matrix product $\hat{\mathbf{A}}\mathbf{X}$ inside the graph convolution layer (7) with certain theoretical guarantees (Chen & Luss, 2018).

Fourth, while we use the same set of hyperparameters for evaluation, nothing prevents problem-specific hyperparameter tuning when one works on an application, specially a challenging one where no general-purpose preconditioners work sufficiently well. We expect that this paper's results based on a naive hyperparameter setting can be quickly improved by the community. Needless to say, improving the neural network architecture can push the success of GNP much farther.

ACKNOWLEDGMENTS

The author would like to thank eight anonymous reviewers whose comments and suggestions help significantly improve this paper.

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

## A  SUPPORTING CODE

The implementation of GNP is available at `https://github.com/jiechenjiechen/GNP`.

## B  FGMRES

FGMRES is summarized in Algorithm 1.

---

**Algorithm 1** FGMRES with $\mathbf{M}$ being a nonlinear operator

---

1: Let $\mathbf{x}_0$ be given. Define $\overline{\mathbf{H}}_m \in \mathbb{R}^{(m+1)\times m}$ and initialize all its entries $h_{ij}$ to zero
2: **loop** until `maxiters` is reached
3:     Compute $\mathbf{r}_0 = \mathbf{b} - \mathbf{A}\mathbf{x}_0$, $\beta = \|\mathbf{r}_0\|_2$, and $\mathbf{v}_1 = \mathbf{r}_0/\beta$
4:     **for** $j = 1, \ldots, m$ **do**
5:         Compute $\mathbf{z}_j = \mathbf{M}(\mathbf{v}_j)$ and $\mathbf{w} = \mathbf{A}\mathbf{z}_j$
6:         **for** $i = 1, \ldots, j$ **do**
7:             Compute $h_{ij} = \mathbf{w}^\top \mathbf{v}_i$ and $\mathbf{w} \leftarrow \mathbf{w} - h_{ij}\mathbf{v}_i$
8:         **end for**
9:         Compute $h_{j+1,j} = \|\mathbf{w}\|_2$ and $\mathbf{v}_{j+1} = \mathbf{w}/h_{j+1,j}$
10:     **end for**
11:     Define $\mathbf{Z}_m = [\mathbf{z}_1, \ldots, \mathbf{z}_m]$ and compute $\mathbf{x}_m = \mathbf{x}_0 + \mathbf{Z}_m \mathbf{y}_m$ where $\mathbf{y}_m = \operatorname{argmin}_{\mathbf{y}} \|\beta \mathbf{e}_1 - \overline{\mathbf{H}}_m \mathbf{y}\|_2$
12:     If $\|\mathbf{b} - \mathbf{A}\mathbf{x}_m\|_2 < \text{tol}$, exit the loop; otherwise, set $\mathbf{x}_0 \leftarrow \mathbf{x}_m$
13: **end loop**

---

## C  PROOFS

*Proof of Theorem 1.* Write $\|\mathbf{r}_m\|_2 \le \|\widetilde{\mathbf{r}}_m\|_2 + \|\mathbf{r}_m - \widetilde{\mathbf{r}}_m\|_2$. It is well known that

$$\|\widetilde{\mathbf{r}}_m\|_2 \le \kappa_2(\mathbf{X})\epsilon^{(m)}(\mathbf{\Lambda})\|\mathbf{r}_0\|_2;$$

see, e.g., Saad (2003, Proposition 6.32). On the other hand, because $\mathbf{r}_m = \mathbf{r}_0 - \mathbf{A}\mathbf{Z}_m\mathbf{y}_m$ where $\mathbf{y}_m$ minimizes $\|\mathbf{r}_0 - \mathbf{A}\mathbf{Z}_m\mathbf{y}\|_2$, we have $\mathbf{r}_m = \mathbf{r}_0 - (\mathbf{A}\mathbf{Z}_m)(\mathbf{A}\mathbf{Z}_m)^+\mathbf{r}_0 = \mathbf{r}_0 - \mathbf{Q}_m\mathbf{Q}_m^\top\mathbf{r}_0$. Therefore, $\|\mathbf{r}_m - \widetilde{\mathbf{r}}_m\|_2 = \|\mathbf{Q}_m\mathbf{Q}_m^\top\mathbf{r}_0 - \widetilde{\mathbf{Q}}_m\widetilde{\mathbf{Q}}_m^\top\mathbf{r}_0\|_2 \le \|\mathbf{Q}_m\mathbf{Q}_m^\top - \widetilde{\mathbf{Q}}_m\widetilde{\mathbf{Q}}_m^\top\|_2\|\mathbf{r}_0\|_2$. $\qquad\square$

*Proof of Corollary 2.* By taking $\widetilde{\mathbf{M}} = \mathbf{A}^{-1}\mathbf{V}_n\mathbf{H}_n\mathbf{V}_n^\top$ and noting that $\mathbf{V}_n^\top = \mathbf{V}_n^{-1}$, we have

$$(\mathbf{V}_n\mathbf{Y})^{-1}\mathbf{A}\widetilde{\mathbf{M}}(\mathbf{V}_n\mathbf{Y}) = \operatorname{diag}(\sigma_1, \ldots, \sigma_n).$$

Then, following Theorem 1, $\|\mathbf{r}_m\|_2 \le \kappa_2(\mathbf{V}_n\mathbf{Y})\epsilon^{(m)}(\mathbf{\Sigma})\|\mathbf{r}_0\|_2$. We conclude the proof by noting that $\kappa_2(\mathbf{V}_n\mathbf{Y}) = \kappa_2(\mathbf{Y})$. $\qquad\square$

## D  APPROXIMATELY EXPONENTIAL CONVERGENCE OF $\|\mathbf{r}_m\|_2$

We could bound the minimax polynomial $\epsilon^{(m)}$ by using Chebyshev polynomials to obtain an (approximately) exponential convergence of the residual norm $\|\mathbf{r}_m\|_2$ for large $m$.

Assume that all the eigenvalues $\sigma_i$ of $\mathbf{\Sigma}$ are enclosed by an ellipse $E(c, d, a)$ which excludes the origin, where $c$ is the center, $d$ is the focal distance, and $a$ is major semi-axis. We have

$$\epsilon^{(m)}(\mathbf{\Sigma}) \le \min_{p \in \mathbb{P}_m,\, p(0)=1} \max_{\sigma \in E(c,d,a)} |p(\sigma)|,$$

by the fact that the maximum modulus of a complex analytical function is reached on the boundary of the domain. Then, we invoke Eqn (6.119) of Saad (2003) and obtain

$$\epsilon^{(m)}(\mathbf{\Sigma}) \le \frac{C_m(a/d)}{C_m(c/d)},$$

where $C_m$ denotes the (complex) Chebyshev polynomial of degree $m$. When $m$ is large,

$$\frac{C_m(a/d)}{C_m(c/d)} \approx \left( \frac{a + \sqrt{a^2 - d^2}}{c + \sqrt{c^2 - d^2}} \right)^m,$$

which gives an (approximately) exponential function in $m$. Applying Corollary 2, we conclude that when $m$ is large,

$$\|\mathbf{r}_m\|_2 \lessapprox \kappa_2(\mathbf{Y}) \|\mathbf{r}_0\|_2 \left( \frac{a + \sqrt{a^2 - d^2}}{c + \sqrt{c^2 - d^2}} \right)^m.$$

## E    SCALABILITY

Based on the architecture outlined in Figure 1, the computational cost of GNP is dominated by matrix-matrix multiplications, where the left matrix is either $\mathbf{A}$ (e.g., in GCONV) or a tall one that has $n$ rows (e.g., in the MLP and Lin layers and in GCONV). This forward analysis holds true for training and inference, because the back propagation also uses these matrix-matrix multiplications. Hence, the computational cost of GNP scales as $O(\mathrm{nz}(\mathbf{A}))$, the same as that of the Krylov solver.

Practical considerations, on the other hand, are more sensitive to the constants hidden inside the big-O notation. For example, one needs to store many (but still a constant number of) vectors of length $n$, due to the batch size, the hidden dimensions of the neural network, and the automatic differentiation. Denote by $c$ this constant number; then, the GPU faces a storage pressure of $\mathrm{nz}(\mathbf{A}) + cn$. Once this amount is beyond the memory capacity of the GPU, one either performs training (and even inference) by using CPU only, offloads some data to the CPU and transfers them back to the GPU on demand, or undertakes more laborious engineering by distributing $\mathbf{A}$ and other data across multiple GPUs. For the last option (multi-GPU and/or distributed training), the literature of GNN training brings in additional techniques (such as sampling and mini-batching the graph) to accelerate the computation (Kaler et al., 2022; 2023).

## F    ADDITIONAL EXPERIMENT RESULTS

### F.1    ADDITIONAL METRICS FOR COMPARING PRECONDITIONERS

In addition to the Iter-AUC and Time-AUC metrics, one may be interested in using the final relative residual norm to compare the preconditioners. To this end, we summarize the results in Figure 7 by using this residual metric, as an extension of Figure 2. Note that for each problem, we solve the linear system twice, once using `rtol` and `maxiters` as the stopping criteria; and the other time using `rtol` and `timeout`. Hence, there are two charts in Figure 7, depending on how the iterations are terminated.

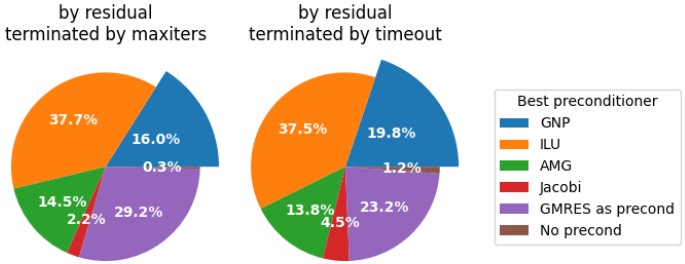

Figure 7: Percentage of problems on which each preconditioner performs the best.

We see that across these two termination scenarios, observations are similar. Compared to the use of the Iter-AUC and Time-AUC metrics, GNP is the best for a similar portion of problems under the residual metric. The portion of problems that ILU and GMRES (as a preconditioner) perform the best increases under this metric, while the portion that AMG performs the best decreases. Jacobi and no preconditioner remain less competitive.

## F.2 PRECONDITIONER CONSTRUCTION TIME AND SOLUTION TIME

In Figure 8, we plot the preconditioner construction time and solution time for each matrix and each preconditioner. This figure is a counterpart of Figure 3 by removing the log scale in the y axis. One sees that the GNP construction time is nearly proportional to the matrix size.

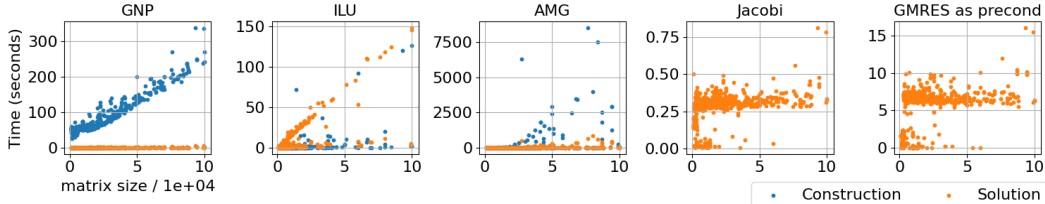

Figure 8: Preconditioner construction time and solution time (using `maxiters` to stop). The y axis is in the linear scale.

## F.3 BREAKDOWN OF BEST PRECONDITIONERS WITH RESPECT TO MATRIX DISTRIBUTIONS

In Figure 9, we plot the proportion of problems on which each preconditioner performs the best, for different matrix sizes, condition numbers, and application areas. This figure is a counterpart of Figure 4. The main observation is that the proportion of problems on which GNP performs the best is relatively flat across matrix sizes. Other observations follow closely from those of Figure 4; namely, GNP is particularly useful for ill-conditioned problems and some application areas (chemical process simulation problems, economic problems, and eigenvalue/model reduction problems).

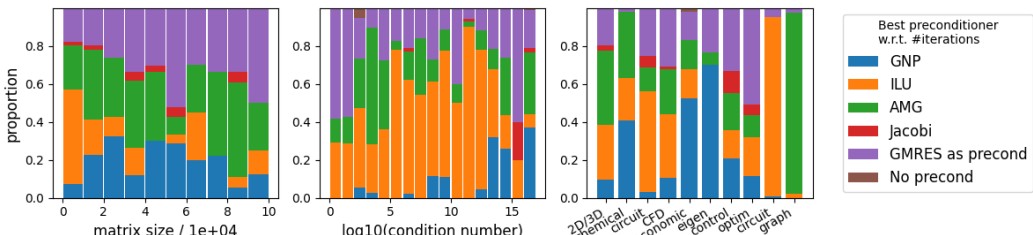

Figure 9: Proportion of problems on which each preconditioner performs the best, for different matrix sizes, condition numbers, and application areas. Only the application areas with the top number of problems are shown. The last bar in the middle plot is for condition number $\geq 10^{16}$. Note that not every matrix has a known condition number.

## F.4 COMPARISON BETWEEN PYAMG AND AMGX

Besides PyAMG, another notable implementation of AMG (which supports using it as a preconditioner and supports GPU) is AmgX https://github.com/NVIDIA/AMGX. Here, we compare the two implementations in terms of robustness and speed.

Unlike PyAMG, which is a blackbox, AmgX leaves the choice of the AMG method and the parameters to the user. We used the driver `examples/amgx_capi.c` and the configuration `src/configs/FGMRES_AGGREGATION.json`, which appears to be the most reasonable for the diverse matrices and application areas at hand. This setting uses aggregation-based AMG rather than classical AMG, which is also consistent with the preference expressed by PyAMG.

Here are some findings.

AmgX is less robust than PyAMG. Table 3 shows that AmgX causes more failures during the construction as well as the solution phase. In fact, there were three cases where AmgX hanged, which hampered automated benchmarking.

Despite a lack of robustness, the construction of AmgX is much faster than that of PyAMG. For AmgX, the distribution of the preconditioner construction time has a minimum 0.006s, median

Table 3: Failures of two implementations of AMG.

|  | PyAMG | AmgX |
|---|---|---|
| Construction failure | 62 (7.15%) | 105 (12.11%) |
| Solution failure | 5 (0.58%) | 36 (4.15%) |

0.017s, and maximum 4.895s. In contrast, the distribution for PyAMG has a minimum 0.005s, median 0.286, and maximum 8505.1s. Nevertheless, there are still 16% of the cases where the PyAMG preconditioner is faster to construct than the AmgX preconditioner.

We skip the comparison of the solution time. This is because in AmgX, the Krylov solver and the preconditioner are bundled inside a C complementation. A fair comparison requires isolating the preconditioner from the solver, because our benchmarking framework implements the solver in Python. However, the isolation is beyond scope due to the complexity of the implementation, such as the use of data structures and cuda handling. Nevertheless, we expect that the solution time of AmgX is faster than that of PyAMG, based on the observations made for preconditioner construction.

### F.5 Changing the Default PyAMG Solver

The blackbox solver of PyAMG, `pyamg.blackbox.solver`, uses the classical-style smoothed aggregation method `pyamg.aggregation.smoothed_aggregation_solver` (SA). For nonsymmetric matrices, however, the AIR method works better (Manteuffel et al., 2019; 2018). In this experiment, we modify the blackbox solver by calling `pyamg.classical.air_solver` for nonsymmetric matrices.

We first tried using the default options of AIR, but encountered "zero diagonal encountered in Jacobi" errors. These errors were caused by the use of `jc_jacobi` as the post-smoother. Then, we replaced the smoothers with the default ones used by SA (`gauss_seidel_nr`). In this case, the solver hung on the problem `FIDAP/ex40`. Before this, 105 problems were solved, among which 83 were nonsymmetric. The results of these 83 systems are summarized in Table 4, suggesting that AIR is indeed better but not always.

Table 4: Counts of problems when comparing AIR with SA for nonsymmetric matrices.

|  | AIR is is better than SA | AIR is worse | The two are equal |
|---|---|---|---|
| By iteration count | 14 | 8 | 61 |
| By final residual | 50 | 29 | 4 |

Overall, we can conclude that AIR has not exhibited its full potential for nonsymmetric matrices. We speculate that a robust implementation of AIR is challenging and that might be the reason why the authors did not use it for the blackbox solver in the first place.

### F.6 Convergence Histories

In Figure 10, we show the convergence histories of the systems appearing in Figure 5, with respect to both iteration count and time. We also plot the training curves. The short solution time of GNP is notable, especially when compared with the solution time of GMRES as a preconditioner. The training loss generally stays on the level of `1e-2` to `1e-3`. For some problems (e.g., `VanVelzen_std1_Jac3`), training does not seem to progress well, but the training loss is already at a satisfactory point in the first place, and hence the preconditioner can be rather effective. We speculate that this behavior is caused by the favorable architectural inductive bias in the GNN.

To provide further examples, we show the convergence histories of all systems in the application areas identified by Figure 4 to favor GNP: chemical process simulation problems (Figures 11–13), economic problems (Figures 14–16), and eigenvalue/model reduction problems (Figures 17–18). In many occasions, when GNP performs the best, it outperforms the second best substantially. It can even decrease the relative residual norm under `rtol` while other preconditioners barely solve the system.

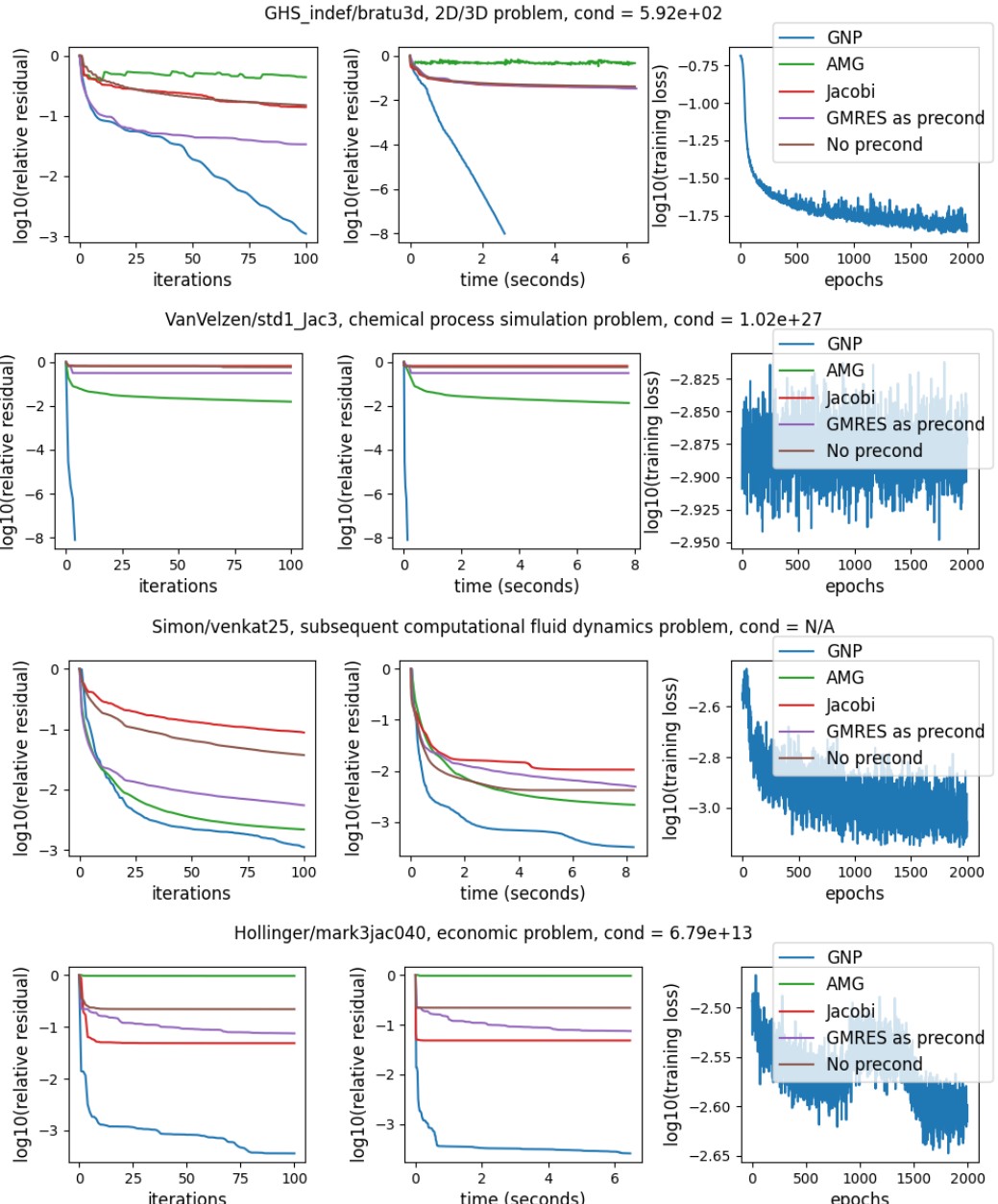

Figure 10: Convergence of the linear system solutions and training history of the preconditioners, for the matrices shown in Figure 5.

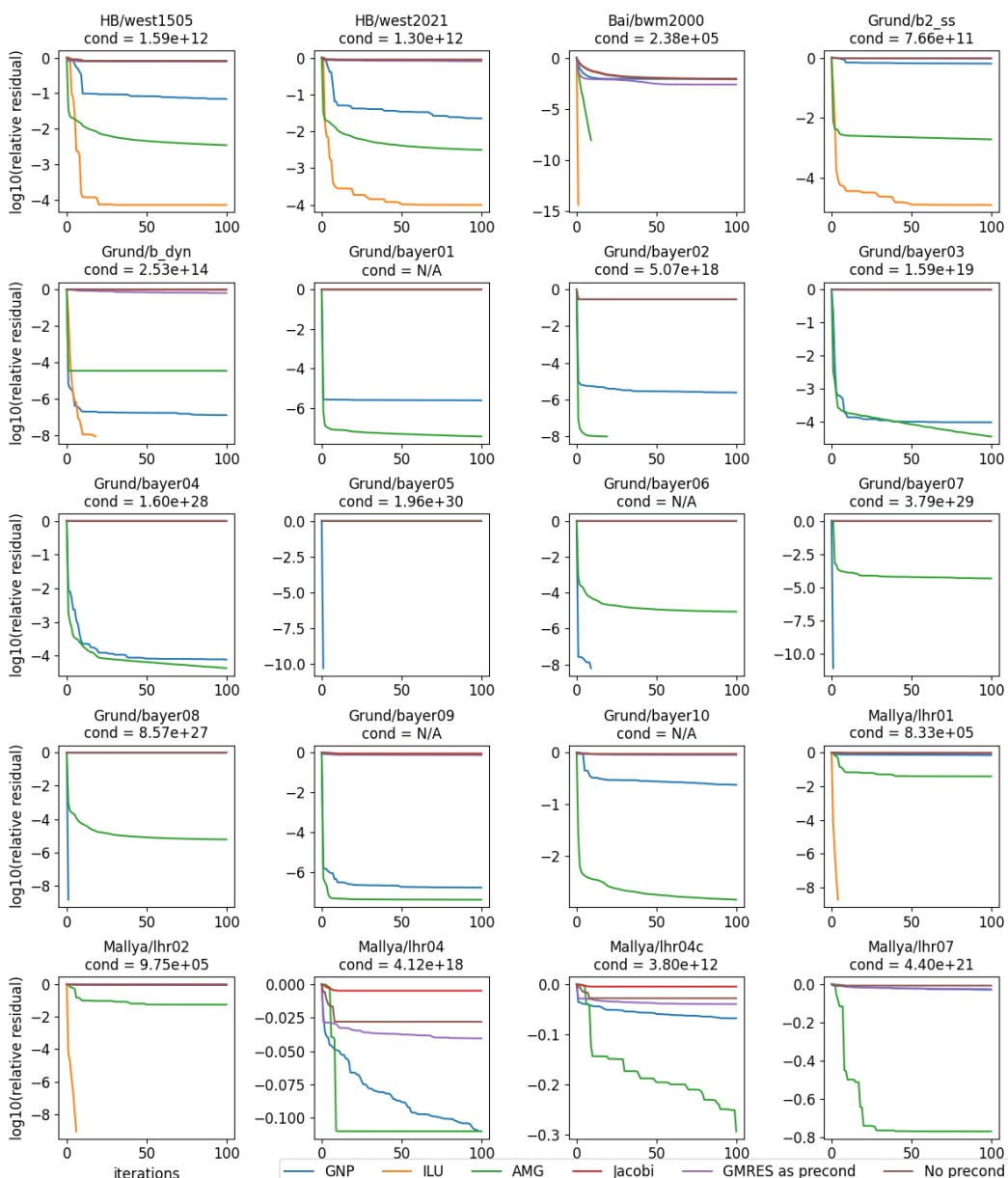

Figure 11: Convergence of the linear system solutions for chemical process simulation problems (1/3).

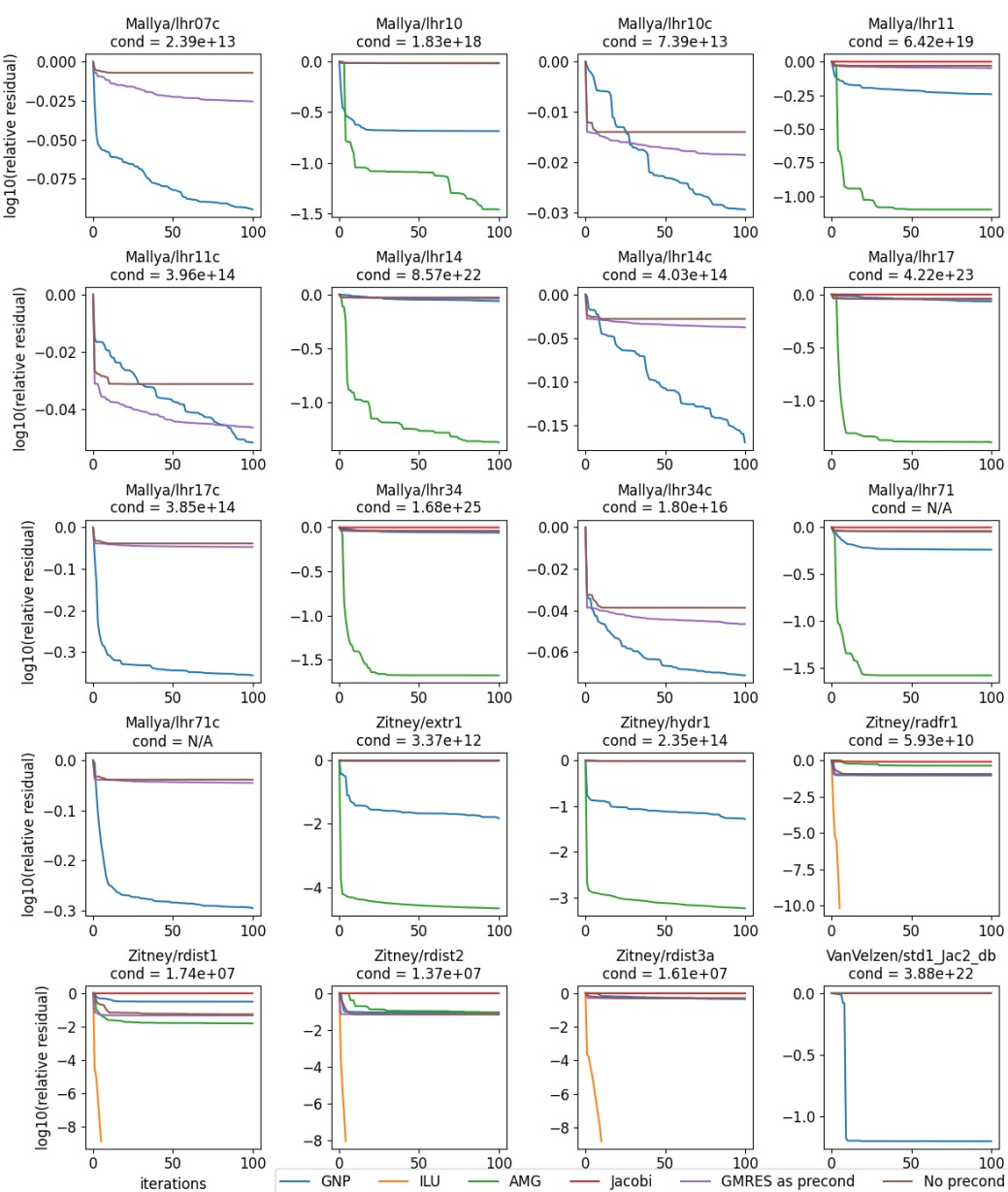

Figure 12: Convergence of the linear system solutions for chemical process simulation problems (2/3).

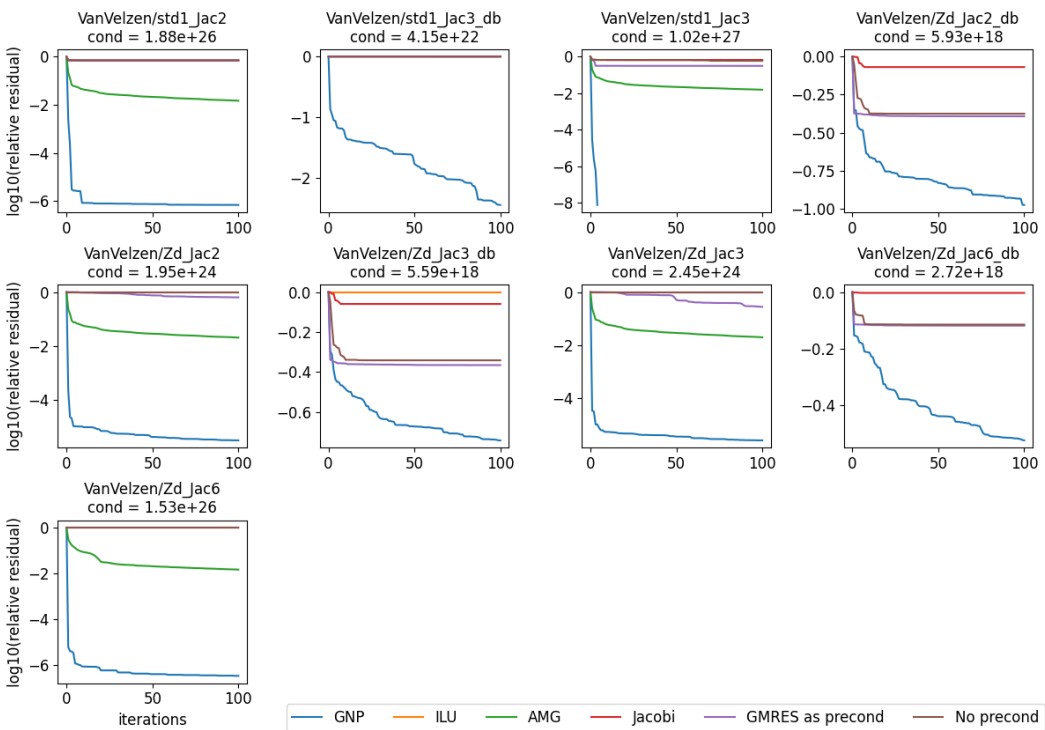

Figure 13: Convergence of the linear system solutions for chemical process simulation problems (3/3).

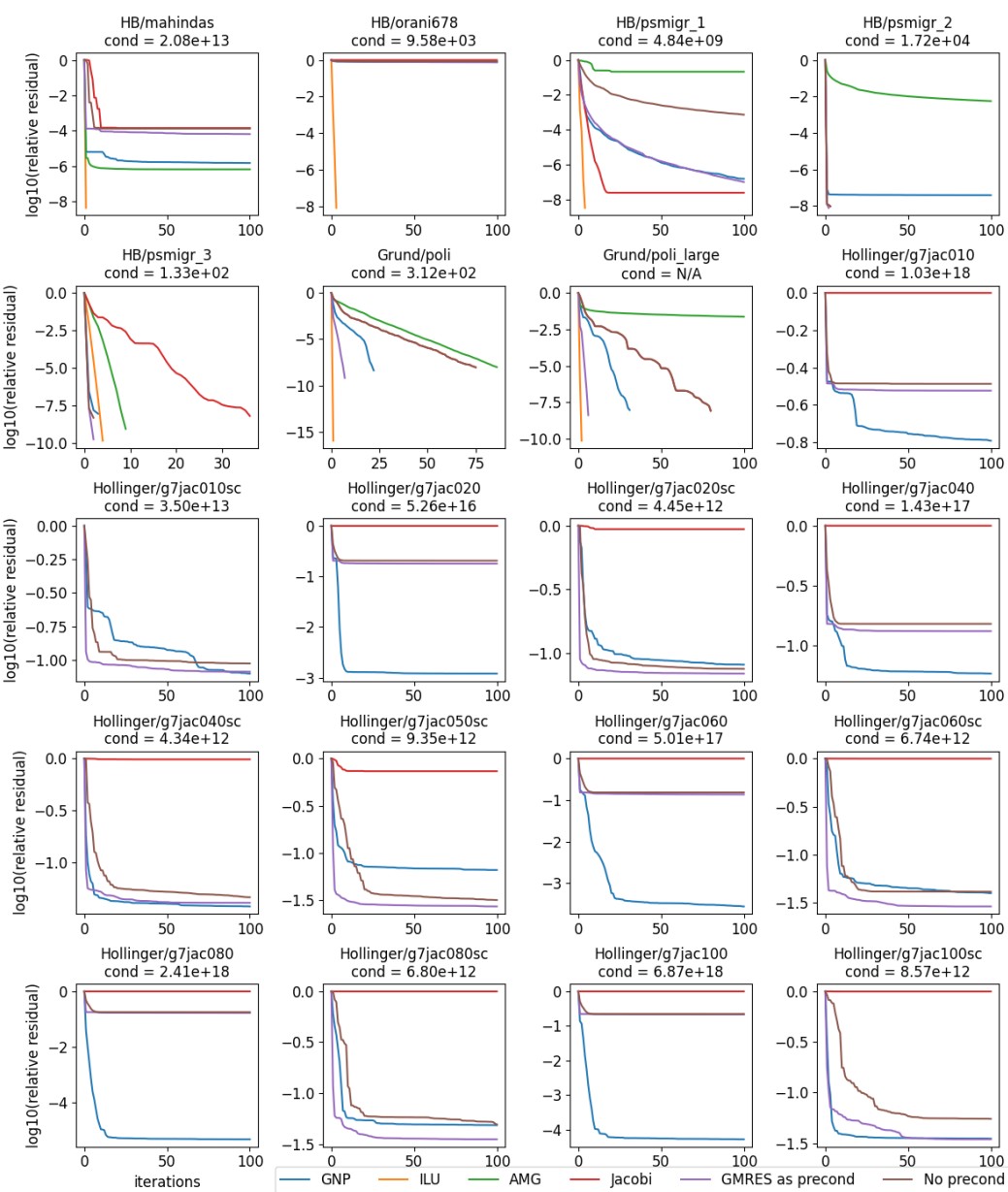

Figure 14: Convergence of the linear system solutions for economic problems (1/3).

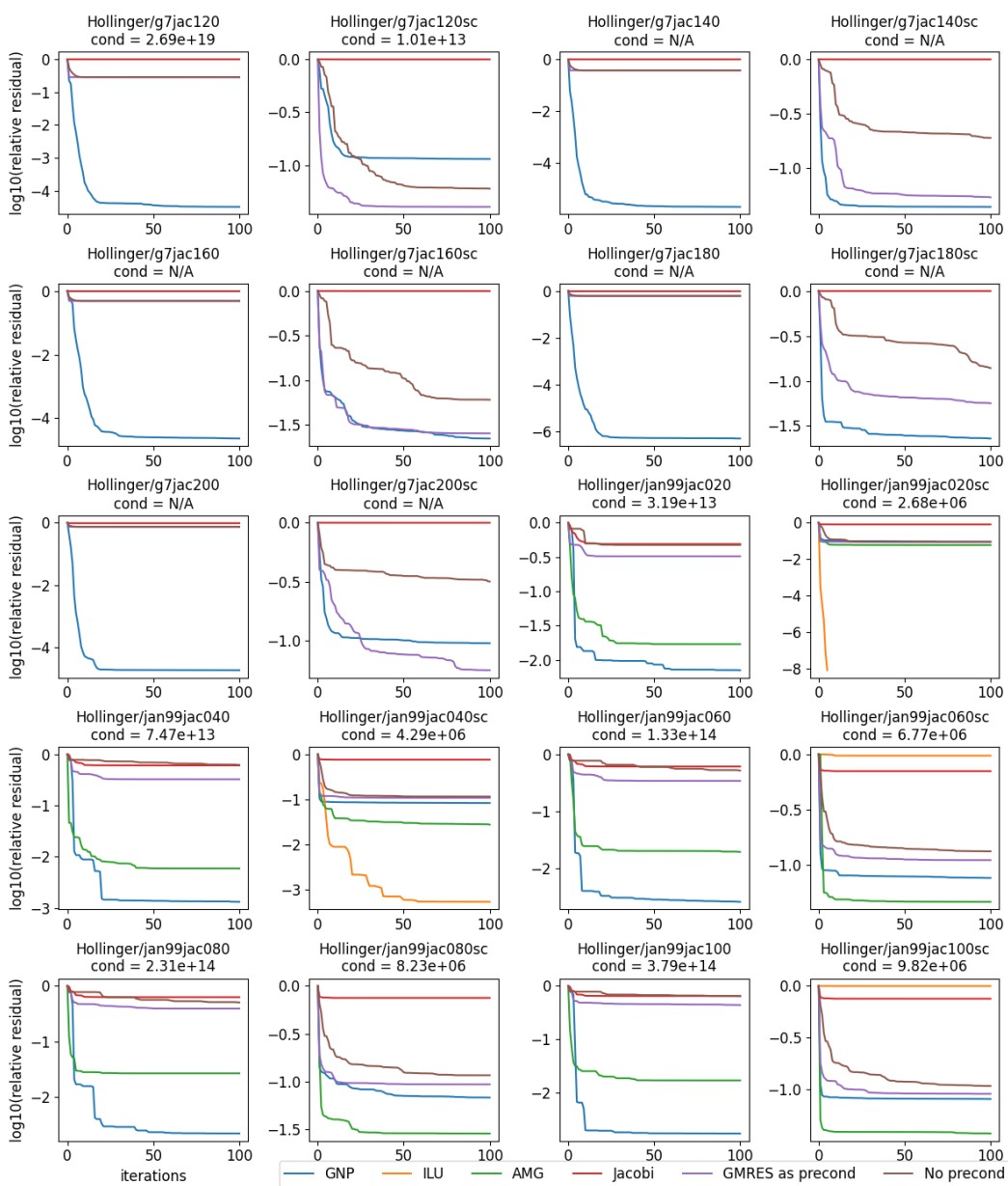

Figure 15: Convergence of the linear system solutions for economic problems (2/3).

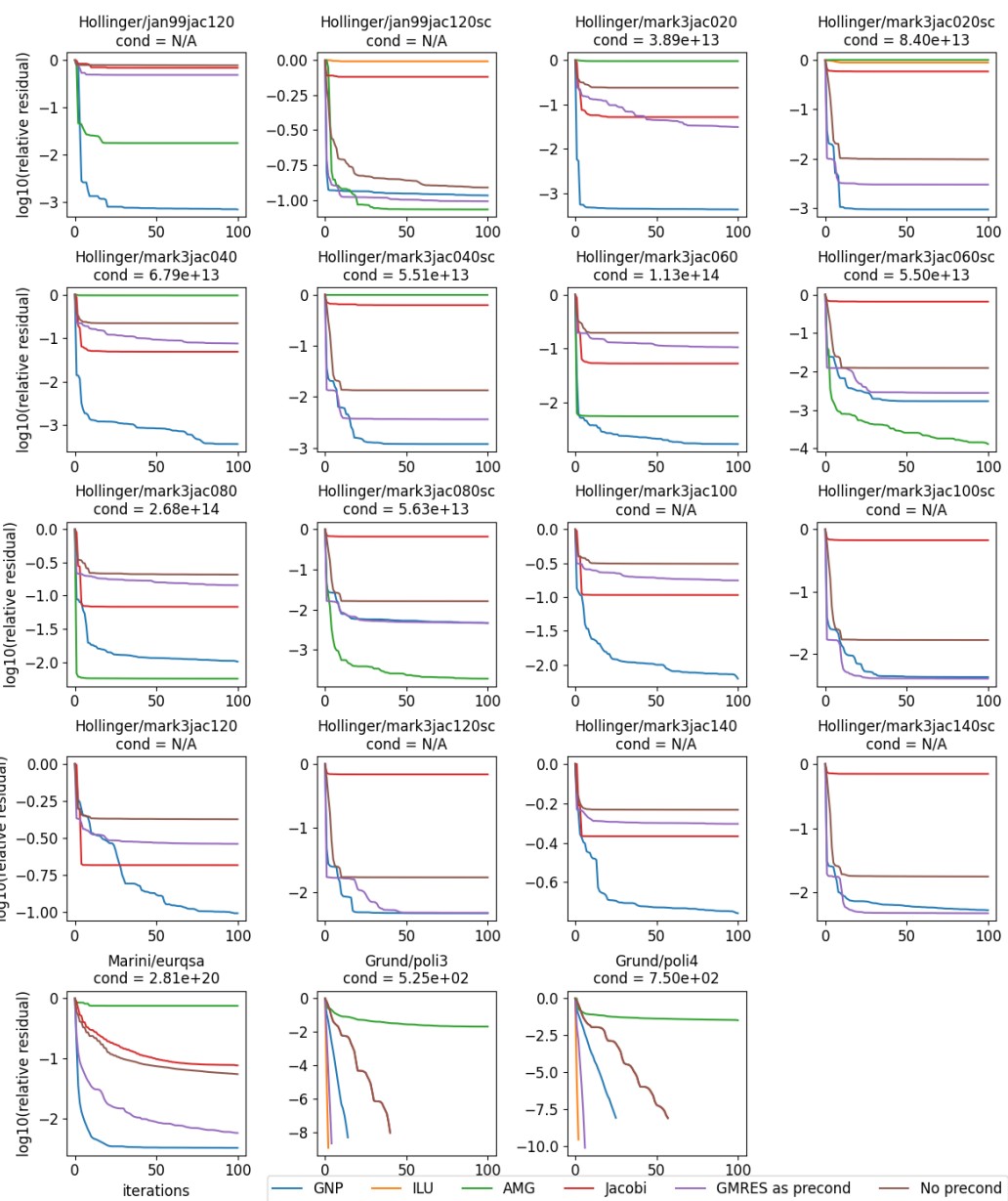

Figure 16: Convergence of the linear system solutions for economic problems (3/3).

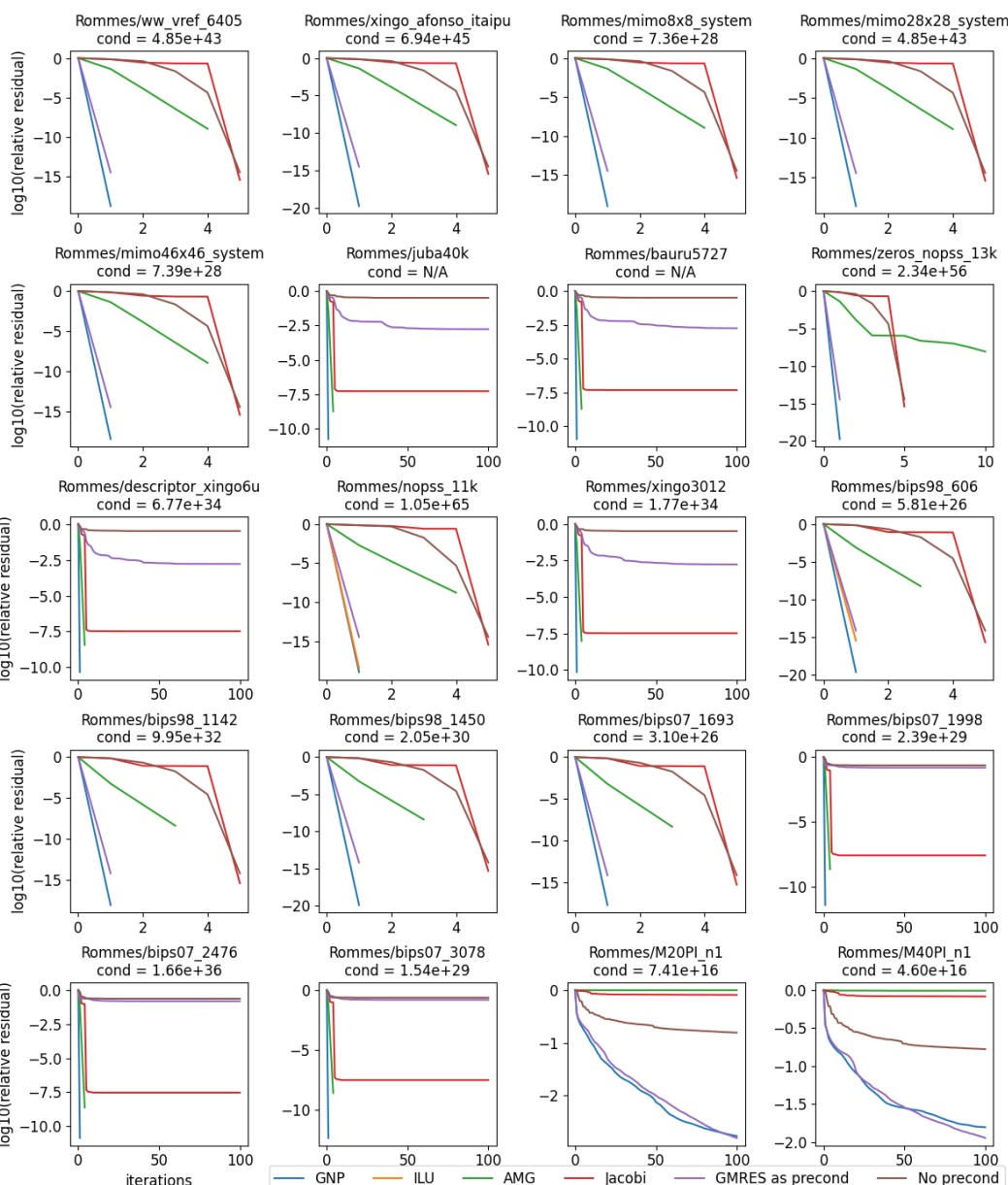

Figure 17: Convergence of the linear system solutions for eigenvalue/model reduction problems (1/2).

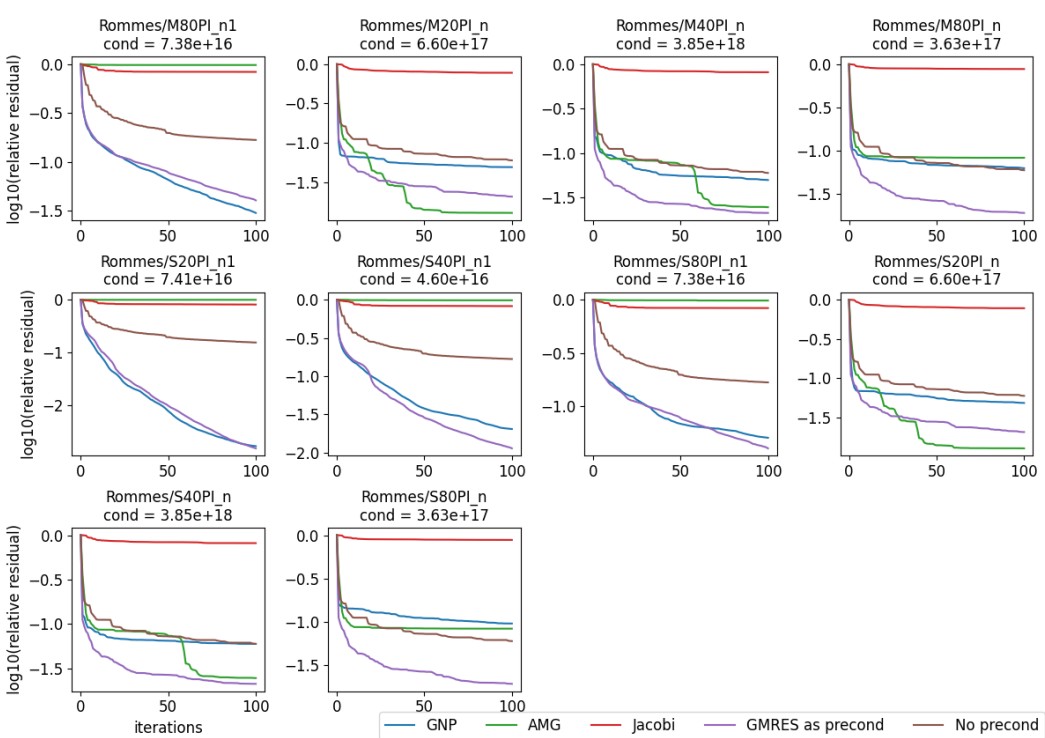

Figure 18: Convergence of the linear system solutions for eigenvalue/model reduction problems (2/2).

