# OpenReview forum: "Graph Neural Preconditioners for Iterative Solutions of Sparse Linear Systems"
_ICLR.cc/2025/Conference — ICLR 2025 Poster_

### Official Review · Reviewer_4sht · 2024-10-27

**Soundness:** 3
**Presentation:** 3
**Contribution:** 2
**Rating:** 6
**Confidence:** 3

**Summary:**

The authors focus on the task of preconditioning linear systems which remains one of the most important problems in scientific computing and applied maths. The authors use a machine learning approach and propose using graph neural networks as a general-purpose preconditioner.

**Strengths:**

- Preconditioning is a relevant problem and discovering new preconditioners for novel applications is resource-demanding
- The paper tackles a large number of test matrices and compares with SOTA black box preconditioners

**Weaknesses:**

- The authors state their result as probably novel. Have they not found anything similar in the literature? I would like them to discuss the difference to the work USING FGMRES TO OBTAIN BACKWARD STABILITY IN MIXED PRECISION by Arioli and Duff, which they have done in the revised version.
- I am not a fan of presenting the results as best performer but it could be that the other methods are really close and this method only outperforms slightly and is terrible for the other problems. I would like to see actual  numbers or convergence plots.

**Questions:**

- For the training data generation at the beginning of 2.2, is this really meaningful? You cannot sample the whole of $\mathbb{R}^n$ to run the problem forwards as also given possible ill-conditioning of $A$ approximating the inverse is difficult. Is this the reason the authors then switch to Arnoldi? This is the basis for many Krylov solvers and it is not clear what the advantage of the machine learning approach his if we have to do Arnoldi anyways?
- How about the sharpness of the Gershgorin estimates? These are not necessarily sharp.
- Do the authors really observe convergence of GMRES in 10 iterations to $10^{-6}$?
- AMG works better for symmetric problems, for nonsymmetric matrices PyAMG comes with better approaches for nonsymmetric problems such as the AIR approach.

---

> ### Author Response · Authors · 2024-11-23
> **Official Comment by Authors (1/2)**
>
> Thank you for your feedback. Please find point-to-point replies below. We also updated the paper with blue text. We will be happy to clarify further if you have additional questions.
>
> **RE: The authors state their result as probably novel. Have they not found anything similar in the literature? I would like them to discuss the difference to the work USING FGMRES TO OBTAIN BACKWARD STABILITY IN MIXED PRECISION by Arioli and Duff.**
>
> Thank you for bringing up the paper by Arioli and Duff (2009). We could comment on the nature of the convergence analysis of this paper and that of ours. Arioli and Duff (2009) are concerned with backward error analysis, which is typically in the following form:
>
> > There exists $\hat{k}$ such that for all $k \ge \hat{k}$, $\|\mathbf{r}_k\| \le c \varepsilon + O(\varepsilon^2)$. Here, $\varepsilon$ is the machine precision, signaling convergence, and $c$ may depend on $n$, $k$, $\|\mathbf{b}\|+\|\mathbf{A}\|\|\mathbf{x}_k\|$, and other terms.
>
> Our analysis, on the other hand, is in the following form:
>
> > There exists $\hat{k}$ such that for all $k \ge \hat{k}$, $\|\mathbf{r}_k\| \lessapprox a \cdot b^k$, where $b<1$ and $a$ is independent of $k$.
>
> The main difference is that our result spells out the convergence rate, while Arioli and Duff (2009) state convergence without a rate. Another difference is that the analysis of Arioli and Duff (2009) assumes finite arithmetics, while our analysis assumes infinite precision.
>
> We have updated the paper with this discussion and removed the novelty claim to avoid confusion, Our novelty is the result but not the act of performing the analysis.
>
> **RE: I am not a fan of presenting the results as best performer but it could be that the other methods are really close and this method only outperforms slightly and is terrible for the other problems. I would like to see actual numbers or convergence plots.**
>
> In Figure 5, we showed the convergence plots for four matrices. In addition, we updated the paper by adding 138 convergence plots at the end (see Appendix Section E.6). These matrices come from application areas identified by Figure 4 to favor GNP: chemical process simulation problems, economic problems, and eigenvalue/model reduction problems. On many occasions, when GNP performs the best, it outperforms the second-best substantially. It can even decrease the relative residual norm under `rtol` while other preconditioners barely solve the system.

---

> > ### Author Response · Authors · 2024-11-23
> > **Official Comment by Authors (2/2)**
> >
> > **RE: For the training data generation at the beginning of 2.2, is this really meaningful? You cannot sample the whole of $\mathbb{R}^n$ to run the problem forwards as also given possible ill-conditioning of $A$ approximating the inverse is difficult. Is this the reason the authors then switch to Arnoldi? This is the basis for many Krylov solvers and it is not clear what the advantage of the machine learning approach his if we have to do Arnoldi anyways?**
> >
> > It is known that the convergence of a Krylov solver to a large degree depends on the eigenvalue distribution, and if there are small eigenvalues, deflating them can substantially help convergence (see Morgan (2002) cited in the paper). Moreover, the approximate eigenvectors do not have to be extremely accurate before the benefits of deflation can be noticed.
> >
> > Following this intuition, we want the preconditioner $\mathbf{M}$ to better approximate (the inverse of) the lower spectrum of $\mathbf{A}$. This can be achieved by using more training data corresponding to the lower spectrum. This training data, in turn, can be generated from the approximate subspace corresponding to the lower Ritz values built by Arnoldi. Eqn (5) spells out the construction of the training data: $\mathbf{x}=\mathbf{V}_m\mathbf{Z}_m\mathbf{S}_m^{-1}\mathbf{\epsilon}$, where $\mathbf{V}_m$ contains the subspace basis and $\mathbf{\epsilon}$ helps sample from a distribution.
> >
> > In Figure 6, we empirically show the benefit of this training distribution. Using x ~ normal leads to lower training losses, while using b ∼ normal leads to higher losses. This is expected, because the former expects similar outputs from the neural network, which is easier to train, while the latter expects drastically different outputs, which is harder to train. Using x from a mixture of the proposed distribution and the normal distribution leads to losses lying in between. Moreover, it leads to the best preconditioning performance, striking a balance between the easiness of neural network training and the goodness of the resulting preconditioner.
> >
> > **RE: How about the sharpness of the Gershgorin estimates? These are not necessarily sharp.**
> >
> > The Gershgorin estimate is indeed not necessarily sharp for the spectral radius. However, it is just a coincidence that the absolute row/column sum has a connection with the spectral radius. Our purpose is mainly to use this sum to normalize $\mathbf{A}$ such that it is neither too small nor too large.
> >
> > **RE: Do the authors really observe convergence of GMRES in 10 iterations to $10^{-6}$?**
> >
> > We are unclear why the reviewer specifically picks the numbers 10 and $10^{-6}$. The convergence of GMRES largely depends on the effectiveness of the preconditioner. In some cases, the residual norm can drop very quickly, such as the use of our preconditioner on `VanVelzen/std1_Jac3`, shown in the second plot of Figure 5.
> >
> > **RE: AMG works better for symmetric problems, for nonsymmetric matrices PyAMG comes with better approaches for nonsymmetric problems such as the AIR approach.**
> >
> > Thank you for bringing the different AMG methods to our attention. We initially used the blackbox solver provided by PyAMG, `pyamg.blackbox.solver`, to avoid the hassle of tuning the solver to each matrix. This solver uses the classical-style smoothed aggregation method `pyamg.aggregation.smoothed_aggregation_solver` (SA). As you suggested, we modified the blackbox solver by calling `pyamg.classical.air_solver` for nonsymmetric matrices.
> >
> > We first tried using the default options of AIR, but encountered "zero diagonal encountered in Jacobi" errors. These errors were caused by the use of `jc_jacobi` as the post-smoother. Then, we replaced the smoothers with the default ones used by SA (`gauss_seidel_nr`). In this case, the solver hung on the problem `FIDAP/ex40`. Before this, 105 problems were solved, among which 83 were nonsymmetric. The results of these 83 systems are summarized in the following table, suggesting that AIR is indeed better but not always.
> >
> > | | AIR is is better than SA | AIR is worse | The two are equal |
> > |-|-|-|-|
> > | By iteration count | 14 | 8 | 61 |
> > | By final residual | 50 | 29 | 4 |
> >
> > Overall, we can conclude that AIR has not exhibited its full potential for nonsymmetric matrices. We speculate that a robust implementation of AIR is challenging and that might be the reason why the authors did not use it for the blackbox solver in the first place.
> >
> > We updated the paper with the above discussions in Appendix Section E.5.
> >
> > Thank you again for your comments. Please let us know if there are other concerns.

---

### Official Review · Reviewer_HJDk · 2024-10-30

**Soundness:** 2
**Presentation:** 3
**Contribution:** 3
**Rating:** 6
**Confidence:** 3

**Summary:**

This paper proposes a GNN-based approach to preconditioning Krylov solvers, offering a novel alternative to traditional algebraic preconditioners. Unlike conventional methods, the GNN-based preconditioner (GNP) approximates the matrix inverse without relying on additional information, making it adaptable and robust across a wide range of matrices. GNP demonstrates predictable training costs, good robustness, and, in some cases, faster convergence than traditional preconditioners like ILU and AMG, which are often hampered by unpredictable construction times and structural limitations.
The contributions:
1. Convergence analysis for FGMRES, a widely used but theoretically underexplored method.
2. New approach to training the neural preconditioner, focusing on effective training data generation by sampling from the bottom eigensubspace of  A to enhance training performance.
3. Scale-equivariant GNN to serve as the preconditioner, addressing the challenge of varying input scales in the training data by enforcing an inductive bias that maintains scale-equivariance.
4. Novel evaluation protocol to test the preconditioner broadly, evaluating across over 800 matrices from 50 diverse application areas in the SuiteSparse collection.

**Strengths:**

1. Introduces a GNN-based preconditioner (GNP) as a novel, general-purpose alternative to traditional methods, approximating the matrix inverse without additional problem-specific information.
2. Offers stable and consistent training times, unlike traditional preconditioners with unpredictable construction times due to matrix irregularities.
3. Contributes a convergence analysis for FGMRES.
4. Tests GNP on over 800 matrices from 50 application areas.

**Weaknesses:**

Sometimes both Iter-AUC and Time-AUC do not give good "weight" for the final residual accuracy. A method that reaches a lower final residual can be underrepresented if it converges more gradually, while a method with fast early convergence but a higher final residual might appear more favorable.

**Questions:**

1. Since neither Iter-AUC nor Time-AUC specifically prioritizes the final residual accuracy, have you considered including a metric that directly measures the final residual? For applications where achieving a specific residual threshold is crucial, this could provide a more balanced evaluation. If not, what are your thoughts on this?
2. Iter-AUC and Time-AUC capture different aspects of convergence (iteration efficiency and time efficiency, respectively). In cases where these metrics might conflict (e.g., one method scores well on Iter-AUC but poorly on Time-AUC), how would you recommend interpreting the results?

---

> ### Author Response · Authors · 2024-11-23
>
> Thank you for your feedback. Please find point-to-point replies below. We also updated the paper with blue text. We will be happy to clarify further if you have additional questions.
>
> **RE: Sometimes both Iter-AUC and Time-AUC do not give good "weight" for the final residual accuracy. A method that reaches a lower final residual can be underrepresented if it converges more gradually, while a method with fast early convergence but a higher final residual might appear more favorable.**
>
> **Since neither Iter-AUC nor Time-AUC specifically prioritizes the final residual accuracy, have you considered including a metric that directly measures the final residual? For applications where achieving a specific residual threshold is crucial, this could provide a more balanced evaluation. If not, what are your thoughts on this?**
>
> Thank you for the thoughtful suggestion. In the updated paper, we summarized the results based on the suggested residual metric in Figure 7 (see Appendix Section E.1). Because for each problem, we solve the linear system twice, once using `rtol` and `maxiters` as the stopping criteria; and the other time using `rtol` and `timeout`. Hence, there are two charts in the figure, depending on how the iterations are terminated.
>
> It is interesting to compare the two charts, as well as to compare them with Figure 2, plotted under the Iter-AUC and Time-AUC metrics. A few observations follow. First, across the two termination scenarios, observations are similar. Second, compared to the use of the Iter-AUC and Time-AUC metrics, GNP is the best for a similar portion of problems under the residual metric. Third, the portion of problems that ILU and GMRES (as a preconditioner) perform the best increases under the residual metric, while the portion that AMG performs the best decreases. Finally, Jacobi and no preconditioner remain less competitive.
>
> **RE: Iter-AUC and Time-AUC capture different aspects of convergence (iteration efficiency and time efficiency, respectively). In cases where these metrics might conflict (e.g., one method scores well on Iter-AUC but poorly on Time-AUC), how would you recommend interpreting the results?**
>
> Indeed, not a single metric paints the full picture. The iteration metric is concerned with the convergence speed (more so from a theoretical standpoint), but if the per-iteration costs are different across methods, it does not properly reflect the time-to-solution progress. This happens in many studies of numerical algorithms, not only in the comparison of two preconditioners but also in the comparison of first-order and second-order optimizers, for example.
>
> Practitioners may care more about time. However, the time metric has its limitations, because in order to measure time, one waits either until the solution reaches a preset residual tolerance, or until the number of iterations max out. For the former, it is impractical to set a universal tolerance for hundreds of problems from diverse domains. For the latter, one solution may max out the iterations early with poor accuracy, while another solution may reach timeout very late but attain good accuracy. Which one is better is arguable. Hence, the time metric is tricky to use.
>
> An easier case is that one evaluates methods on only one (or a few) problem(s). In this case, it is possible to set a problem-specific residual tolerance. Then, one prioritizes the time metric over the iteration metric.
>
> Thank you again for your comments. Please let us know if there are other concerns.

---

> ### Comment · Reviewer_HJDk · 2024-11-25
>
> Dear authors, thank you for your detailed responses, espesially for adding residual metric and analyzing it. Now my comments are addressed. However, I believe my assessment is already correct.

---

### Official Review · Reviewer_7HSM · 2024-11-03

**Soundness:** 3
**Presentation:** 3
**Contribution:** 2
**Rating:** 3
**Confidence:** 4

**Summary:**

The paper proposes a novel approach using Graph Neural Networks (GNNs) as general-purpose preconditioners for solving large, sparse linear systems. The authors present an empirical evaluation of their method, showing that it outperforms traditional preconditioning techniques like ILU and AMG in terms of construction time and execution efficiency, especially for ill-conditioned matrices. The method is tested on a diverse set of over 800 matrices, suggesting significant advantages in robustness and performance compared to existing methods.

**Strengths:**

- **Innovative Approach:** The use of GNNs as preconditioners is a fresh perspective that leverages recent advancements in machine learning.

- **Empirical Validation:** The extensive evaluation on a wide range of matrices provides strong evidence of the proposed method's effectiveness across various problem domains.

- **Robustness:** The proposed method demonstrates high robustness with a low failure rate compared to traditional preconditioners, particularly in handling ill-conditioned problems.

- **Theoretical Contributions:** The paper includes a convergence analysis for the flexible GMRES method, which adds valuable theoretical insights to the practical implementation.

**Weaknesses:**

- **Scalability Concerns:** While the paper mentions robustness, it lacks detailed discussions on scalability, particularly regarding the performance of GNNs with significantly larger matrices.

- **Dependence on Training Data:** The effectiveness of the GNN-based preconditioner seems to heavily rely on the quality and diversity of the training data. The paper does not explore the implications of this dependence thoroughly.

- **Limited Theoretical Foundation:** Although the empirical results are compelling, the theoretical grounding could be strengthened, especially concerning the convergence properties of the proposed methods under varied conditions.

**Questions:**

1. How does the proposed preconditioner handle matrices outside the tested conditions, particularly in real-world applications?

2. Could you elaborate on the training data generation process? How might biases in the training set affect the performance of the GNN?

3. What are the implications of your findings for future research in preconditioning techniques, especially concerning adapting the GNN for a wider array of problems?

---

> ### Author Response · Authors · 2024-11-23
> **Official Comment by Authors (1/2)**
>
> Thank you for your feedback. Please find point-to-point replies below. We also updated the paper with blue text. We will be happy to clarify further if you have additional questions.
>
> **RE: Scalability Concerns: While the paper mentions robustness, it lacks detailed discussions on scalability, particularly regarding the performance of GNNs with significantly larger matrices.**
>
> The computational cost of GNP is dominated by matrix-matrix multiplications, where the left matrix is either $\mathbf{A}$ or a tall one that has $n$ rows. This forward analysis holds for training and inference, because the back propagation also uses these matrix-matrix multiplications. Hence, the computational cost of GNP scales as $O(nnz(\mathbf{A}))$, the same as that of the Krylov solver.
>
> Practical considerations, on the other hand, are more sensitive to the constants hidden inside the big-O notation. For example, one needs to store many (but still a constant number of) vectors of length $n$, due to the batch size, the hidden dimensions of the neural network, and the automatic differentiation. Denote by $c$ this constant number; then, the GPU faces a storage pressure of $nnz(\mathbf{A}) + cn$. Once this amount is beyond the memory capacity of the GPU, one either performs training (and even inference) by using CPU only, offloads some data to the CPU and transfers them back to the GPU on demand, or undertakes more laborious engineering by distributing $\mathbf{A}$ and other data across multiple GPUs.
>
> We have included this discussion in Appendix D of the updated paper.
>
> **RE: Dependence on Training Data: The effectiveness of the GNN-based preconditioner seems to heavily rely on the quality and diversity of the training data. The paper does not explore the implications of this dependence thoroughly.**
>
> In Figure 6, we compared three approaches of generating the training data, including two naive choices ($\mathbf{x}$ is drawn from standard normal and $\mathbf{b}$ is drawn from standard normal) and our proposed design (denoted by "x ~ mix"). We compare their performance in terms of training quality and preconditioner quality. In terms of training quality, the figure suggests that using x ~ normal leads to lower losses, while using b ∼ normal leads to higher losses. This is expected, because the former expects similar outputs from the neural network, which is easier to train, while the latter expects drastically different outputs, which is harder to train. Using x from the proposed mixture leads to losses lying in between.
>
> More important is the preconditioner quality. Our design leads to the best preconditioning performance. In other words, our design strikes the best balance between the easiness of neural network training and goodness of the resulting preconditioner.
>
> **RE: Limited Theoretical Foundation: Although the empirical results are compelling, the theoretical grounding could be strengthened, especially concerning the convergence properties of the proposed methods under varied conditions.**
>
> In Corollary 2 and Appendix Section C, we established the exponential convergence of the residual norm $\| \mathbf{r}_m \|_2$ with respect to the number of iterations $m$, namely
> $$
> \| \mathbf{r}_m \|_2 \lessapprox
> \kappa_2(\mathbf{Y}) \| \mathbf{r}_0 \|_2 \left( \frac{a + \sqrt{a^2-d^2}}{c + \sqrt{c^2-d^2}} \right)^m.
> $$
> This convergence is based on mild assumptions (i.e., FGMRES does not incur breakdown and the resulting upper Hessenberg matrix can be diagonalized).

---

> ### Author Response · Authors · 2024-11-23
> **Official Comment by Authors (2/2)**
>
> **RE: How does the proposed preconditioner handle matrices outside the tested conditions, particularly in real-world applications?**
>
> The proposed preconditioner is learned separately for each matrix. The learning is based on a set of $(\mathbf{b},\mathbf{x})$ data pairs sampled from a distribution. The actual tested condition, $\mathbf{x}=\mathbf{1}$, does not come from this distribution. Hence, our evaluation can test out-of-distribution generalization.
>
> For a new problem, one would train a preconditioner for it by sampling fresh $(\mathbf{b},\mathbf{x})$ data pairs consistent with the problem dimension.
>
> Our tested problems come from SuiteSparse, which collects real-world matrices from numerous applications. For example, the HB collection, from which 30 matrices we tested, "stem from actual applications and exhibit numerical pathologies that arise in practice." [1] Testing 867 matrices from 50 applications in the experiments allows us to draw observations based on a large population. This is way more comprehensive than existing preconditioner papers that focus on one or two problems only.
>
> Reference:
>
> [1] I. S. Duff, Roger G. Grimes, and John G. Lewis. Sparse matrix test problems. ACM Trans. Math. Softw. 15(1), 1--14. 1989.
>
> **RE: Could you elaborate on the training data generation process? How might biases in the training set affect the performance of the GNN?**
>
> The training data is generated in a streaming fashion, meaning that it is not a finite dataset where a data point will be reused in each epoch. Each batch contains 16 $(\mathbf{b},\mathbf{x})$ pairs, where 8 of them have $\mathbf{x}$ generated from the standard normal and the rest of them have $\mathbf{x}$ generated from $\mathbf{V}_m\mathbf{Z}_m\mathbf{S}_m^{-1}\mathbf{\epsilon}$, with $\mathbf{\epsilon}$ being standard normal, $\mathbf{Z}_m$ and $\mathbf{S}_m$ from the SVD of $\overline{\mathbf{H}}_m$, and $\overline{\mathbf{H}}_m$ and $\mathbf{V}_m$ from the Arnoldi process. The Arnoldi process and the SVD are computed only once before all the batches. The vector $\mathbf{b}$ is computed as $\mathbf{A}\mathbf{x}$.
>
> We are unsure what you meant by "biases in the training set". We do not introduce bias in the training data. Happy to elaborate further if you could clarify.
>
> **RE: What are the implications of your findings for future research in preconditioning techniques, especially concerning adapting the GNN for a wider array of problems?**
>
> Our findings reassert the folklore wisdom that there is not a one-size-fits-all preconditioner, especially when one moves beyond solving PDEs and enters the territories of data science and machine learning (for example, some of our test matrices come from statistical problems, graph problems, and optimization problems). Traditionally, practitioners choose the preconditioner for a particular problem by experience; trials and errors are an integral part of their work. Our preconditioner offers an additional choice to them. Our experiments suggest that a substantial portion of problems benefit from this choice.
>
> We consider benchmarking to be an innovative contribution of this work, as there is rarely a preconditioner paper testing the proposed technique on a large number of problems and painting a full picture of its applicability. A benefit of large-scale benchmarking is that there is little room for problem-specific tuning. Hence, we offer default parameters, which a practitioner can try first. From there, the practitioner can tune the parameters and even the neural architecture for a particular problem.
>
> For future research, we believe that split preconditioners, sequential fine-tuning, and distributed implementation are the imminent follow-up work, as laid out in the final section of the paper. These directions extend this work to problems that are SPD, that are time-varying, and that are exa-scale, respectively. The empirical success of our findings should also encourage the exploration of universal approximation of GNNs for matrix function (e.g., matrix inverse) times a vector.
>
> Thank you again for your comments. Please let us know if the rebuttal and the update of the paper clear your concerns.

---

### Official Review · Reviewer_VMvP · 2024-11-04

**Soundness:** 4
**Presentation:** 4
**Contribution:** 3
**Rating:** 6
**Confidence:** 5

**Summary:**

Authors propose use Graph Neural Network (GNN) as a general-purpose preconditioner. They offer a convergence analysis for flexible GMRES. A new effective training data generation is proposed to training GNN. They develop a scale-equivariant GNN as the preconditioner. A novel evaluation protocol proposed by authors is being for general-purpose preconditioners.

**Strengths:**

* Authors pay their attention to the problem which is linked to using neural networks (in particular, GNN) as the preconditioner. In this sense, the preconditioner is a nonlinear operator and, using a standard preconditioned solver with this preconditioner is not correct. In this case, the convergence theory is broken. They concentrate on flexible GMRES and present an original convergence analysis.
* Authors propose original scheme of data generation. Suggested sampling is linked with eigen-subspace of the $\mathbf{A}$.
* Of course, the idea of using GNN as the preconditioner is not novel. Authors offer a fresh approach to normalize $\mathbf{A}$. This prevents the potential division-by-zero issue that can arise in the standard normalization of GCN. The major innovation of the architecture is its scale-equivariance, i.e. the input space of the neural network is restricted.
* Proposed architecture is checked on the SuiteSparse matrix collection https://sparse.tamu.edu (square, real-valued 867 matrices (not spd-matrices) from 50 application areas, whose count of non-zero elements is less than 2M. To compare with classical methods such as ILU, AMG, and GMRES author propose two novel metrics.

**Weaknesses:**

* Authors use $\ell_1$ residual norm as the training loss. Is it possible to use $\ell_2$ norm or others as the training loss?
* Authors compare their architecture with classical methods(GNP, ILU, AMG, Jacobi, and GMREs). It will be good to compare their approach and metrics with the existing general-purpose preconditioners which are used GNN as the preconditioner. See for example, https://proceedings.mlr.press/v202/li23e/li23e.pdf, https://arxiv.org/pdf/2405.15557.
* Are there exist some constraints to use this approach for SPD-matrices?
* Why is not the full GitHub repository made available for review?
* Why do the authors only assume the ground truth solution $\mathbf{x} = 1$?

**Questions:**

See weaknesses.

---

> ### Author Response · Authors · 2024-11-23
>
> Thank you for your feedback. Please find point-to-point replies below. We also updated the paper with blue text. We will be happy to clarify further if you have additional questions.
>
> **RE: Authors use $\ell_1$ residual norm as the training loss. Is it possible to use $\ell_2$ norm or others as the training loss?**
>
> We mostly follow the general practice of robust regression, as the $\ell_1$ loss is less susceptible to outliers. Using other loss functions, such as the $\ell_2$ loss, is possible. In preliminary experiments, we did not find a substantial difference between the use of $\ell_1$ versus $\ell_2$. We speculate that this is because the signal $\mathbf{A}\mathbf{x}$ follows a Gaussian distribution and thus rarely incurs outliers. We have clarified the rationale of using $\ell_1$ in the updated paper.
>
> **RE: Authors compare their architecture with classical methods (GNP, ILU, AMG, Jacobi, and GMREs). It will be good to compare their approach and metrics with the existing general-purpose preconditioners which are used GNN as the preconditioner.**
>
> Existing GNN approaches for learning general-purpose preconditioners include the cited works by Li et al. (2023), Häusner et al. (2023), Bånkestad et al. (2024), and your suggested reference Trifonov et al. (2024). They are inapplicable to our scenario for two reasons. First, they learn a neural network that generates a preconditioner for SPD matrices (because, e.g., they learn the incomplete Cholesky factor), whereas in our work, we are concerned with non-SPD matrices only (please see more discussions in the next response item regarding the separate treatment of the SPD case). Second, the cited works learn the neural network from a distribution of matrices parameterized by the given problem (e.g., a parameterized PDE, or a PDE with different boundary conditions or mesh discretizations). In our case, we are given an algebraic matrix $\mathbf{A}$ without the problem parameterization.
>
> We have included your suggested reference in the updated paper.
>
> **RE: Are there exist some constraints to use this approach for SPD-matrices?**
>
> Both (F)GMRES and the proposed GNP preconditioner apply to all matrices, including SPD ones. However, there are theoretical and practical advantages to using the alternative CG solver for SPD matrices, because in this case, the Arnoldi process is equivalent to the Lanczos process, which results in a tridiagonal matrix and gets rid of orthogonalization in practice. In this case, an SPD preconditioner works better with CG. Hence, we discuss in the final section that an extension of this work is to develop GNNs and training techniques that lead to a self-adjoint operator (the operator version of SPD preconditioner).
>
> **RE: Why is not the full GitHub repository made available for review?**
>
> Thank you for your interest in the code. Indeed, we plan to organize the code, package the preconditioner as a python module, and release the benchmarking framework when this paper is accepted for publication. By releasing the code, we hope to encourage the field to adopt the preconditioner and build their evaluation on the benchmarking framework.
>
> **RE: Why do the authors only assume the ground truth solution $\mathbf{x}=\mathbf{1}$?**
>
> Setting $\mathbf{x}=\mathbf{1}$ is a common practice in testing linear solvers and preconditioners, as is used in, for example, the original FGMRES paper. We deliberately avoid setting $\mathbf{x}$ to be Gaussian random, because this (partially) matches the training data generation, resulting in the inability to test out-of-distribution cases. We have updated the paper with this rationale.
>
> Thank you again for your comments. Please let us know if there are other concerns.

---

### Meta-Review · Area_Chair_B8cS · 2024-12-18

**Metareview:**

This work proposes using graph neural networks (GNNs) as general-purpose preconditioners for solving large, sparse linear systems, particularly for challenging ill-conditioned matrices. Empirical evaluation on 800+ matrices shows that GNN-based preconditioners offer faster and more predictable construction times compared to traditional methods like ILU and AMG, with faster execution times than Krylov-based preconditioners. The approach has strong potential for large-scale problems in various fields, including PDEs, economics, and optimization.

Three reviewers recommended acceptance, emphasizing the substantial contributions of the paper. They highlighted the soundness and novelty of the proposed preconditioning technique, noting its fresh perspective. Another reviewer praised the thoroughness of the experiments, which evaluate performance across 800 matrices from 50 application areas, effectively showcasing the method's capabilities.
However, one reviewer recommended rejection, citing concerns about scalability, reliance on training data, and a limited theoretical foundation.

The AC sides with the majority vote, acknowledging that the paper's strengths significantly outweigh its weaknesses.

**Additional Comments On Reviewer Discussion:**

All reviewers raised some technical questions, which the authors have addressed satisfactorily. I strongly recommend that the authors incorporate these remarks in the final version.

---

### Decision · Program_Chairs · 2025-01-22

Accept (Poster)